# Time Series Reasoning via Process-Verifiable Thinking Data Synthesis and Scheduling for Tailored LLM Reasoning

**Jiahui Zhou** [1]  **Dan Li** [1]  **Boxin Li** [2]  **Xiao Zhang** [2]  **Erli Meng** [2]  **Lin Li** [1]  **Zhuomin Chen** [1]
**Jian Lou** [1]  **See-Kiong Ng** [3]

## Abstract

Time series is a pervasive data type across various application domains, rendering the reasonable solving of diverse time series tasks a long-standing goal. Recent advances in large language models (LLMs), especially their reasoning abilities unlocked through reinforcement learning (RL), have opened new opportunities for tackling tasks with long Chain-of-Thought (CoT) reasoning. However, leveraging LLM reasoning for time series remains in its infancy, hindered by the absence of carefully curated time series CoT data for training, limited data efficiency caused by underexplored data scheduling, and the lack of RL algorithms tailored for exploiting such time series CoT data. In this paper, we introduce VeriTime, a framework that tailors LLMs for time series reasoning through data synthesis, data scheduling, and RL training. First, we propose a data synthesis pipeline that constructs a time series–text multimodal dataset with process-verifiable annotations. Second, we design a data scheduling mechanism that arranges training samples according to a principled hierarchy of difficulty and task taxonomy. Third, we develop a two-stage reinforcement finetuning featuring fine-grained, multi-objective rewards that leverage verifiable process-level CoT data. Extensive experiments show that VeriTime substantially boosts LLM performance across diverse time series reasoning tasks. Notably, it enables compact 3B–4B models to achieve reasoning capabilities on par with or exceeding those of larger proprietary LLMs. The code for VeriTime is available at https://github.com/ZhoujhZoe/VeriTime.

[1]Sun Yat-sen University [2]Xiaomi Corporation [3]National University of Singapore. Correspondence to: Dan Li <lidan263@mail.sysu.edu.cn>.

*Proceedings of the 43rd International Conference on Machine Learning*, Seoul, South Korea. PMLR 306, 2026. Copyright 2026 by the author(s).

## 1. Introduction

Time series (TS) data is a fundamental modality pervasive across diverse domains, including natural phenomena, energy, finance, healthcare, transportation, and industry (Shen et al., 2025; Xu et al., 2026a; Zhong et al., 2025; Hu et al., 2026). With the advent of Large Language models (LLMs) and their success in natural language processing and beyond, researchers have recently shown growing interest in exploiting LLMs as a unified solution for tackling diverse TS tasks, such as forecasting (Chen et al., 2025; Lyu et al., 2026; Masserano et al., 2025), anomaly detection (Li et al., 2025; Park et al., 2025b; Xu et al., 2026b), classification (Huang et al., 2025; Goswami et al., 2024; Hossen et al., 2026), and imputation (Fons et al., 2025; Yang et al., 2025b).

A prevailing approach to integrate LLMs with time series tasks utilizes supervised fine-tuning (SFT) on TS-text datasets (Wang et al., 2025a; Niu et al., 2025a). For instance, ChatTS (Xie et al., 2025) and Time-MQA (Kong et al., 2025) fine-tune models respectively using synthetic and real-world Q&A pairs. However, these approaches primarily focus on aligning time series representations with LLMs, lacking the explicit consideration (e.g., chain-of-thought and reinforcement learning) for complex reasoning.

Accordingly, the critical research gap lies in how to adequately reflect the intricate time series patterns and fully integrate them into LLM reasoning processes (Wang et al., 2026; Guo et al., 2026; Zhang et al., 2026). Real-world time series inherently convey complex multi-scale temporal correlations and causal dynamics (Jang et al., 2025; Liu et al., 2025c), making it crucial for models to explicitly comprehend the underlying relational structures before forming rationales and conclusions. However, most existing work adopts basic prompting techniques to interpret TS sequences and feeds observed patterns to LLM directly (Liu et al., 2024b; Merrill et al., 2024; Gruver et al., 2023). Inspired by the impressive reasoning performance of frontier models (Jaech et al., 2026; Guo et al., 2025), the reinforcement learning (RL) paradigm has recently emerged as a nascent yet promising direction to further enhance LLM time series reasoning capabilities (Zhang et al., 2025; Manfredi et al., 2026). LangTime (Niu et al., 2025b) integrates cross-

domain pretraining with temporal comprehension prompts, and introduces TimePPO to optimize forecasting. Guan et al. (2026) propose TimeOmni, a unified model trained on the human-annotated TSR-Suite using SFT and task-grounded RL. However, these approaches relies heavily on outcome-based rewards, overlooking the process-level signals necessary to verify intermediate reasoning steps. Overall, the direction of empowering LLMs for time series reasoning remains in its infancy and warrants further investigation into the validity and interpretability of the reasoning process.

Another critical gap is the lack of tailored time series reasoning data that are extensive enough to cover diverse representative time series tasks (Jing et al., 2026; Gonen et al., 2025). Existing benchmarks are limited by inadequate task diversity, unclear calibration standards, and many rely predominantly on simplified synthetic data only that cannot adequately reflect real-world complexity and variability (Kong et al., 2026; Wang & Zhao, 2024). Therefore, this calls for a dual approach that leverages synthetic data to ensure coverage across diverse scenarios, while utilizing real-world data for knowledge-intensive domains. MTBench (Chen et al., 2026) reveals that LLMs consistently struggle with tasks demanding nuanced temporal understanding and effective multimodal integration. This underscores the necessity of constructing a high-quality benchmark characterized by diverse hybrid data sources, tailored Chain-of-Thought reasoning paths, and process-verifiable signals to rigorously supervise reasoning trajectories (Rousseau et al., 2025).

In this work, we propose **VeriTime**, a time series reasoning framework that enhances LLMs' TS reasoning capabilities through process-verifiable reasoning data synthesis and tailored RL effectively exploiting time series-specific intermediate learning signals from the generated data.

**Time Series Reasoning Data Synthesis.** We introduce Time Series Reasoning Generation (TSRgen) to generate stepwise time series reasoning data, forming TSRBench. TSRBench comprises time series-text Q&A pairs covering both scenario-based and knowledge-based tasks, and spanning synthetic and real-world time series. With a TS-specific CoT generation strategy that captures key aspects of process-level performance across diverse TS tasks, TSRBench integrates valid reasoning trajectories with process-verifiable annotations to support both training and evaluation.

**Tailored RL for Time Series Reasoning.** Next, we introduce a two-stage Reinforcement Fine-Tuning (RFT) framework to enhance LLMs' TS reasoning capabilities by effectively exploiting TSRBench. The first supervised fine-tuning stage leverages the TS-tailored CoT paradigm to learn from the reasoning trajectories with successful task completion, while the second RL stage refines the model's reasoning performance. Rather than merely optimizing for final prediction accuracy, we design fine-grained, multi-objective

rewards that incorporate process-level reasoning signals to explicitly assess the validity and logical coherence of intermediate TS reasoning steps. Furthermore, to improve training data efficiency, we devise a data scheduling mechanism for RFT via a selective rollout strategy that dynamically identifies learnable and representative samples best suited to the corresponding RFT stage based on task difficulty and model performance.

Our main contributions are summarized as follows:

- We propose the TSRgen pipeline that generates high-quality TS-text reasoning data tailored for enhancing LLMs' ability on time series reasoning tasks. The tailored reasoning data is automatically curated based on both synthetic and real-world time series data and covers seven tasks. To the best of our knowledge, it is *the first time series reasoning dataset that integrates TS-tailored CoT reasoning paths with process-verifiable annotations*.

- We propose a two-stage Reinforcement Fine-Tuning (RFT) framework coming with fine-grained and multi-objective rewards that explicitly evaluate the validity of intermediate reasoning steps across diverse TS tasks. A data scheduling mechanism is further introduced to improve the efficiency and effectiveness of RFT.

- Extensive experiments demonstrate that the generated time series reasoning data and the proposed VeriTime framework significantly enhance LLMs' time series reasoning capabilities with an average improvement of over 35% across tasks, highlighting the importance of supervising intermediate reasoning processes.

## 2. Related Work

### 2.1. LLM Reasoning for Time Series

Early attempts to explore the potentials of LLMs for time series (TS) primarily rely on prompting-based methods (Tang et al., 2025; Liu et al., 2025a; Niu et al., 2025a). For instance, PromptCast (Xue & Salim, 2024) and LLMTime (Gruver et al., 2023) demonstrates that LLMs can perform zero-shot time series extrapolation by treating TS values as text tokens. Despite their effectiveness, plain prompting techniques inadequately elicit LLMs' reasoning ability, as general linguistic knowledge often fails to align with TS-specific features and patterns (Park et al., 2025a). Merrill et al. (2024) further find that current LLMs exhibit unsatisfactory reasoning ability on TS data, reporting that human annotators outperform LLMs by over 30% in etiological reasoning tasks, with advanced models such as GPT-4 achieving only marginal improvements. To bridge this gap, another line of work employs supervised fine-tuning (SFT) to enhance TS reasoning capabilities of LLMs (Liu et al., 2024c;

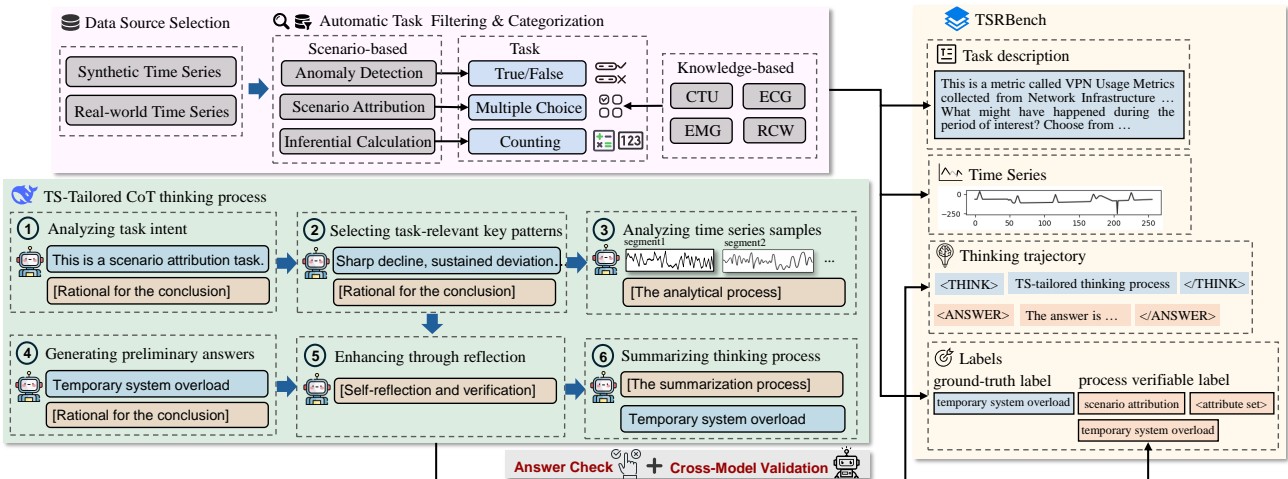

*Figure 1.* The overall framework of the time series reasoning data generation pipeline *TSRgen*.

Kong et al., 2026). Furthermore, some approaches focus on the task-specific adaptation of LLM backbones for diverse time series applications (Cheng et al., 2025; Ning et al., 2025), where GPT4TS (Zhou et al., 2023) adapts language model backbones to TS tasks via fine-tuning.

Reinforcement learning (RL) (Guo et al., 2025; Team et al., 2025) has emerged as a promising paradigm for enhancing LLM reasoning. To meet the growing demand for reliable time series reasoning, several studies have explored applying RL to time series domains (Niu et al., 2025b; Liu et al., 2025d). For example, Guan et al. (2026) find that RL performs reliably when the LLM is anchored with fundamental temporal priors. Zhou et al. (2026) propose a two-stage RL framework for multi-step reasoning in time series forecasting. Further broadening the scope, TimeMaster (Zhang et al., 2025) introduces an RL-based approach that enables multimodal LLMs to perform structured and interpretable reasoning over visualized time series inputs. Despite these advances, existing RL-based methods in the TS domain primarily optimize for final prediction accuracy, while overlooking the evaluation of reasoning processes validity, structural coherence and logical consistency.

### 2.2. Time Series Reasoning Benchmark

Benchmarking LLM reasoning over time series has attracted growing attention with datasets spanning diverse domains and task types (Qiao et al., 2026; Qiu et al., 2025). Some work construct synthetic datasets to target specific reasoning objectives. For instance, Zhou & Yu (2025) introduce four anomaly detection datasets that emulate distinct time series patterns and abnormal behaviors, while Merrill et al. (2024) create multiple-choice benchmarks based on GPT4-generated descriptions. However, synthetic datasets alone cannot fully reflect LLMs' reasoning performance under realistic conditions. To address this issue, multimodal bench-

marks such as MTBench (Chen et al., 2026) and Time-Bench (Liu et al., 2025d) integrate textual and temporal information, spanning domains including finance, weather, and news. They primarily assess long-text understanding and treat time series only as an auxiliary modality.

Several studies shifts toward richer reasoning evaluation over real-world data. Specifically, TSAIA (Ye et al., 2025) and TSQA (Kong et al., 2025) shift toward evaluating compositional reasoning and multi-step analysis through forecasting, anomaly detection, and open-ended Q&A tasks. Taking a further step, RATs40K (Yang et al., 2026) is a multimodal benchmark explicitly annotated for anomaly reasoning. Overall, existing datasets mainly emphasize the correctness of final predictions and lack mechanisms to verify or interpret intermediate reasoning processes. Moreover, there is still no unified and generalizable framework for advancing TS reasoning in a principled manner.

## 3. Methodology

### 3.1. TSRgen: Reasoning Data Synthesis Pipeline

We propose an automated Time Series Reasoning generation (TSRgen) pipeline to construct a high-quality time series-text multimodal reasoning dataset, as depicted in Figure 1. TSRgen generates process-verifiable reasoning data based on both synthetic and real-world TS across diverse scenarios by integrating a rule-based extractor with LLM excelling at complex reasoning (e.g., DeepSeek-R1) guided by well-designed instructions. It begins with time series filtering and task categorization, followed by the TS-tailored Chain-of-Thought (CoT) to generate structured reasoning trajectories with correctness verification. Finally, TSRgen aligns and consolidates the corresponding time series samples, task descriptions, reasoning trajectories, and process-verifiable labels into a unified dataset *TSRBench*.

### 3.1.1. DATA AND TASK SELECTION

To enhance LLM's time series reasoning capability, both **scenario-based** and **knowledge-based** reasoning tasks are included in the TSRBench dataset. Scenario-based tasks are synthesized from ChatTS (Xie et al., 2025), where reasoning tasks are generated under predefined conditions. The original data is filtered and reorganized into three distinct task categories, each representing a specific reasoning style: (1) *Anomaly Detection* (deductive reasoning), (2) *Scenario Attribution* (causal reasoning), and (3) *Inferential Calculation* (quantitative reasoning). On the other hand, knowledge-based tasks are derived from real-world datasets across medical (Clifford et al., 2017), healthcare (Reaz et al., 2006), energy, and bioacoustics (Abousleiman et al., 2013) domains. Overall, TSRBench includes true/false, multiple-choice, and open-ended Q&A formats. More details are in Appendix B.

### 3.1.2. TS-TAILORED CoT THINKING PROCESS

Previous work shows that naive NLP reasoning techniques are less effective for time series reasoning tasks (Liu et al., 2025b; Merrill et al., 2024), making it non-trivial to leverage LLMs for this domain. Effective TS reasoning hinges on the thinking trajectory that aligns with time series properties and validates intermediate steps. To guide LLMs toward structured and interpretable time series reasoning, TSRgen constructs a TS-tailored reasoning process with six logically ordered steps illustrated in Figure 1. It first discerns **task intent** and **key attributes** (e.g., trends, thresholds, domain-specific terms), then isolates **critical segments** for analysis. Building upon insights from the preceding steps, the LLM drafts a **preliminary answer**, performs **backtracking** and **self-reflection** (Ji et al., 2023; Tyen et al., 2024) to ensure comprehensiveness and eliminate interfering factors, and finally **summarizes** the reasoning trajectory to ensure the response is coherent and traceable.

### 3.1.3. AUTOMATIC REASONING DATA ANNOTATION AND VERIFICATION

Based on the predefined TS-tailored thinking process, the TSRgen then crafts time series reasoning questions and generates comprehensive reasoning trajectories by leveraging reasoning LLMs to develop structured and interpretable reasoning patterns. By assessing both the correctness of final predictions and the validity of intermediate reasoning steps, TSRgen further verified the quality of the TSRBench. The key steps of dataset construction are summarized as follows.

**Automatic Task-Specific Instance Selection.** TSRgen employs a rule-based extractor to filter original instances that satisfy predefined reasoning criteria (scenario-based and knowledge-based) to improve annotation quality. A task-specific keyword set is defined to automate the selection process for each reasoning task. For anomaly detection, for

instance, TS-text pairs containing terms such as *anomalous*, *anomalies*, or *extreme* are selected.

**Reasoning Procedure Assessment.** Generating high-quality Chain-of-Thought (CoT) reasoning paths necessitates that the TS-tailored thinking strategy is both comprehensive and logically sound. To this end, TSRgen leverages DeepSeek-R1, offering exceptional general reasoning capabilities, as the expert model for generating reasoning trajectories. DeepSeek-R1 is instructed with curated TSRBench questions and guided to follow our predefined TS-tailored CoT steps. TSRgen automatically filters out cases where the LLM's final answer aligns with the ground-truth label, as these instances serve as the basis for verifying the completeness and validity of intermediate reasoning steps.

**Verifiable Process Annotation Extraction.** To facilitate Reinforcement Learning (RL) for improving LLM time series reasoning performance, both the final label and the effective supervision of intermediate reasoning steps are crucial. TSRgen first conducts cross-LLM validation to evaluate the quality of reasoning steps (details presented in subsection B.2), and then utilizes a rule-based annotation extractor to derive verifiable process-level labels from the reasoning chains generated by DeepSeek-R1. It targets key reasoning steps within the TS-tailored thinking process that can be independently verified and used to support validation during RL training.

Specifically, for task intent understanding (**Step 1**), TSRgen extracts the target entity as the verifiable label. For the key attributes extraction **Step 2**, TSRgen constructs a task-specific attribute set by integrating attributes identified by DeepSeek-R1 with target patterns extracted from the ChatTS dataset under a GPT selector. For preliminary answer generation and summarization (**Steps 4 and 6**), process-level labels are derived from expert predictions and validated against ground-truth annotations. Evaluation is omitted for critical segment analysis and self-reflection steps to enable exploration flexibility.

In conclusion, the TSRgen is an automated TS-text dataset synthesis pipeline that integrates diverse reasoning tasks within a structured, TS–tailored reasoning dataset, named as TSRBench. Unlike existing time series reasoning benchmarks that overlook the validity and interpretability of intermediate processes during LLM training and evaluation, to the best of the authors' knowledge, TSRBench is the first time series reasoning dataset with verifiable multi-step thinking process annotations.

### 3.2. VeriTime: RL Fine-tuning with Data Scheduling

Post-training for LLMs via RL typically requires substantial computational resources with large-scale training data. Prior studies have shown that, during the RL phase of reasoning LLMs, the quality of training data often outweighs its quan-

*Figure 2.* Overview of the proposed VeriTime. It consists of three stages: (1) Stage 1 leverages TSRBench to warmup a base LLM $\theta_0$ into $\theta_1$, which is subsequently used to perform difficulty stratification over all TSRBench tasks. (2) Stage 2 fine-tunes $\theta_1$ on samples with normal difficulty to obtain $\theta_2$, equipping the model with TS-oriented CoT thinking paradigms. (3) Stage 3 applies RL to optimize the accuracy of both final predictions and key intermediate reasoning steps, further enhancing reasoning quality and yielding $\theta_{\text{final}}$.

tity in determining model performance (Wang et al., 2025c; Zheng et al., 2025). However, in the context of time series reasoning, guidance on selecting suitable data for reinforcement fine-tuning (RFT) remains limited (Tian et al., 2026). To address this gap, we propose a selective RFT data rollout strategy that identifies learnable and representative samples based on the difficulty hierarchy of TSRBench, enhancing both efficiency and effectiveness in RL-based post-training. The overview of VeriTime is depicted in Figure 2.

**Problem Definition.** Formally, we define the training dataset $\mathcal{D}_{train}$ as a collection of $n$ instances:

$$\mathcal{D}_{train} = \{(x_i, y_i, \mathcal{L}_i) \mid i = 1, 2, \ldots, n\}, \quad (1)$$

where each instance consists of a time series reasoning problem $x_i$, its corresponding Chain-of-Thought reasoning path $y_i$, and a ground-truth label set $\mathcal{L}_i$. The label set $\mathcal{L}_i = \{l_{\text{final}}, l_{\text{process}}\}$ comprises a final ground-truth label and a set of process-supervision labels. For each case, the model learns to map $x_i \rightarrow (y_i, \mathcal{L}_i)$ through the reasoning process.

**Difficulty Hierarchy.** We first categorize sample difficulty based on the task taxonomy of TSRBench. Knowledge-based tasks $\mathcal{D}_{\text{knowledge}}$ involve longer temporal sequences and are curated from real-world professional domains, where time series patterns and interdependencies are inherently more complex and diverse than those of synthetic datasets, whose patterns are generated under simplified and scenario-specific conditions. This distinction is further supported by the performance of general LLMs, as depicted in Table 2 and Table 3. Hence, we classify $\mathcal{D}_{\text{knowledge}}$ as hard samples ($\mathcal{D}_{hard}$), while scenario-based tasks $\mathcal{D}_{\text{scenario}}$ are further partitioned based on the following procedure:

**Warmup Training.** We first initialize the model's reasoning capability via supervised fine-tuning on a small randomly sampled subset $\mathcal{D}_{\text{sample}} \subset \mathcal{D}_{\text{train}}$ (10% of total data). The base model $\pi_{\theta_0}$ is fine-tuned on $\mathcal{D}_{sample}$ through supervised learning to obtain the warmed-up model $\pi_{\theta_1}$:

$$\theta_1 = \arg\min_\theta \frac{1}{|\mathcal{D}_{\text{sample}}|} \sum_{(x_i, y_i) \in \mathcal{D}_{\text{sample}}} \mathcal{L}_{\text{SFT}}(\pi_\theta(x_i), y_i), \quad (2)$$

where $\mathcal{L}_{\text{SFT}}$ denotes the supervised fine-tuning loss. This stage enables the model to learn the basic reasoning format and structure of time series reasoning problems.

**Data Selection.** Next, the warmed-up model $\pi_{\theta_1}$ performs inference on $\mathcal{D}_{\text{scenario}}$. Each instance is evaluated based on the correctness of the ground-truth label $l_{\text{final}}$, and is partitioned into a normal subset $\mathcal{D}_{\text{normal}}$ and a hard subset $\mathcal{D}_{\text{hard}}$ accordingly:

$$\begin{aligned} \mathcal{D}_{\text{normal}} &= (x_i, y_i, \mathcal{L}_i) \in \mathcal{D}_{\text{train}} \mid \text{Acc}(f_{\theta_1}(x_i), l_{\text{final}}) = 1, \\ \mathcal{D}_{\text{hard}} &= (x_i, y_i, \mathcal{L}_i) \in \mathcal{D}_{\text{train}} \mid \text{Acc}(f_{\theta_1}(x_i), l_{\text{final}}) = 0. \end{aligned} \quad (3)$$

This step enables the model to identify normal learnable samples and challenging samples based on its reasoning performance, without excessive computational cost.

**Two-stage RFT Training.** The training process consists of two stages, corresponding to the two subsets defined above. First, we fine-tune the model $\pi_{\theta_1}$ on $\mathcal{D}_{normal}$ to enhance stability and general reasoning capability:

$$\theta_2 = \arg\min_\theta \frac{1}{|\mathcal{D}_{\text{normal}}|} \sum_{(x_i, y_i) \in \mathcal{D}_{\text{normal}}} \mathcal{L}_{\text{SFT}}(\pi_\theta(x_i), y_i). \quad (4)$$

Subsequently, to further enhance reasoning performance, we apply reinforcement learning using the GRPO algorithm (Shao et al., 2024) on the hard subset $\mathcal{D}_{hard}$:

$$\theta_{\text{final}} = \arg\max_\theta \mathbb{E}_{(x_i, \mathcal{L}_i) \sim \mathcal{D}_{\text{hard}}}\Big[ \mathcal{R}(\pi_\theta(x_i), \mathcal{L}_i) \Big], \quad (5)$$

where $\mathcal{R}(\cdot)$ is the reward function measuring consistency with both the final label and intermediate process labels.

Overall, the hierarchical process enables the model to enhance its reasoning ability in a progressive manner. It first consolidates learnable patterns and then focuses on challenging cases through targeted RL.

### 3.3. Multi-Objective Reward Design of VeriTime

To enhance LLM time series reasoning capabilities, it is essential to ensure correct final predictions while optimizing the intermediate reasoning trajectory when interpreting task-specific time series. This requires simultaneously balancing structural discipline, coherent multi-step reasoning, and final-answer accuracy. VeriTime therefore adopts fine-grained reward components for multi-objective optimization, categorized into structural rewards $r_{\text{struct}}$, the hard reward $r_{\text{hard}}$, and process rewards $r_{\text{process}}$.

*Table 1.* Data Statistics of TSRBench.

| Category | Scenario-based Reasoning | | | | Knowledge-based Reasoning | | | | |
|---|---|---|---|---|---|---|---|---|---|
| Task | Total | Anomaly Detection | Scenario Attribution | Inferential Calculation | Total | CTU | ECG | EMG | RCW |
| # Samples | 2520 | 1180 | 930 | 410 | 1820 | 270 | 780 | 450 | 320 |
| # Avg. Time Points | - | 300 | 300 | 324 | - | 720 | 500 | 600 | 500 |
| # Avg. Token Count | 2784 | 2759 | 2932 | 2783 | 4627 | 5942 | 4239 | 4990 | 4201 |

To encourage well-structured and complete reasoning, we design two **structural reward** components: the **format reward** $r_{\text{fmt}}$ enforces adherence to the required output template, while the **length reward** $r_{\text{len}}$ encourages responses of appropriate detail and depth, with penalties for deviations from the desired range, as defined by the minimum required length $L_{\text{thr}}$ and the upper tolerance threshold $L_{\text{tol}}$. Furthermore, the **hard reward** $r_{\text{hard}}$ evaluates the factual correctness of the predicted final answer $A(y)$, serving as the primary optimization criterion with the highest magnitude. Detailed formulations are summarized in Appendix C.

Reliable intermediate reasoning steps are crucial for interpretable time series reasoning. To comprehensively assess the quality of the LLM reasoning trajectory, we further introduce four **process-level rewards** aligned with the TS-tailored reasoning steps.

**Task Comprehension Reward.** $r_{\text{intent}}$ evaluates whether LLM correctly grasps the core objective in its initial judgment $J_1(y)$. A fuzzy similarity score with threshold $T_1$ determines correctness.

$$r_{\text{intent}}(y) = \mathbb{I}\left[\text{FuzzySim}\big(J_1(y), \hat{J}_1\big) \geq T_1\right]. \quad (6)$$

**Critical Pattern Reward.** $r_{\text{pattern}}$ measures the model's capability to identify essential reasoning patterns for task resolution. Given a ground-truth set $S = \{s_1, s_2, ..., s_m\}$ and extract the model-identified patterns from $J_2(y)$ into a candidate set $D = \{d_1, d_2, ..., d_n\}$. Matches are counted if the maximum similarity with elements in $S$ exceeds the threshold $T_2$. Pattern identification is important but less critical than final answer correctness, with rewards of $+2$ compared to $+5$ in hard reward.

$$M(y) = \sum_{d \in D} \mathbb{I}\left[\max_{s \in S} \text{FuzzySim}(d, s) \geq T_2\right] \quad (7)$$

$$r_{\text{pattern}}(y) = \begin{cases} +2.0, & M(y) \geq 2, \\ +1.0, & M(y) = 1, \\ +0.0, & \text{otherwise.} \end{cases} \quad (8)$$

**Answer Alignment Reward.** $r_{\text{align}}$ checks if the model's first concrete answer in $J_4(y)$, which is produced after primary reasoning steps, aligns with the ground truth. As process rewards emphasize the validity of intermediate results, correct alignment is strongly rewarded to encourage coherent reasoning, whereas misalignments are not penalized

to maintain exploratory flexibility. With ExactMatch$(u, v)$ return 1 if $u = v$ and 0 otherwise:

$$r_{\text{align}}(y) = 2.0 \times \mathbb{I}\left[\text{ExactMatch}\big(J_4(y), \hat{a}\big)\right]. \quad (9)$$

**Answer Verification Reward.** Following backtracking and self-reflection in $J_5(y)$, the model produces a final, verified decision $J_6(y)$. $r_{\text{verify}}$ verifies its consistency with the ground truth $\hat{a}$, which reflects the model's self-assessment ability and commitment to a validated conclusion.

$$r_{\text{verify}}(y) = \mathbb{I}\left[\text{ExactMatch}\big(J_6(y), \hat{a}\big)\right]. \quad (10)$$

The overall reward aggregates all components uniformly, as their role differences are reflected in the reward score ranges during the design of each individual reward.

## 4. Evaluation

In this section, we present a comprehensive evaluation of the proposed VeriTime framework. Experiments are designed to address the following Research Questions (RQs):

**RQ1:** How effectively does VeriTime improve LLM performance across diverse reasoning tasks in TSRBench?
**RQ2:** How does TSRgen's time series-tailored Chain-of-Thought enhance LLM's timer series reasoning capabilities?
**RQ3:** How does the multi-objective reward design contribute to the step-wise performance of VeriTime?
**RQ4:** How does the data scheduling strategy affect the trade-off between performance and efficiency?

### 4.1. Experimental Setup

**Datasets.** TSRBench supports both training and evaluation of time series reasoning models, comprising two types of tasks: *scenario-based reasoning*, where each case is situated in a realistic scenario paired with corresponding synthetic data, and *knowledge-based reasoning*, constructed from real-world datasets. The scale and statistics of each task is summarized in Table 1. For scenario-based tasks, the training-to-test ratio is approximately 5:1, with instances randomly sampled under the TSRBench curation pipeline. For knowledge-based reasoning tasks, the training–test split follows the original partitioning of the respective source datasets. Additionally, we evaluate VeriTime on two other open-source benchmarks, TimeSeriesExam (Cai et al., 2024) and DROP (Dua et al., 2019).

**Models and Baselines.** We select two moderate-sized LLMs, Qwen2.5-3B-Instruct and Qwen3-4B-Instruct (Yang et al., 2025a), for training and validation of VeriTime. For comparison, we include baseline LLMs of comparable parameter scales from mainstream series such as Llama, Mistral, and GPT. Additionally, we evaluate time series language models Time-MQA (Kong et al., 2025) and Time-R1 (Zhou et al., 2026). For knowledge-based tasks, we also evaluate six representative traditional TS models.

*Table 2.* Comparison of Accuracy (%) across three scenario-based reasoning tasks.

| | Overall | Anomaly Detection | Scenario Attribution | Inferential Calculation |
|---|---|---|---|---|
| *General LLMs* | | | | |
| GPT-5 | 83.49 | 90.56 | 78.98 | **80.95** |
| Gemini-3-Pro | 78.91 | 75.28 | 81.82 | 79.63 |
| DeepSeek-R1-Distill-Qwen-7B | 52.93 | 48.33 | 59.66 | 49.52 |
| Qwen2.5-7B-instruct | 66.81 | 70.56 | 69.89 | 55.24 |
| Meta-Llama3-8B-Instruct | 59.22 | 68.89 | 64.77 | 33.33 |
| Mistral-7B-v0.3 | 59.22 | 75.00 | 61.36 | 28.57 |
| GPT-4o-mini | 70.43 | 82.12 | 61.36 | 65.71 |
| *Time Series Language Models* | | | | |
| Time-MQA (Qwen2.5-7B) | 41.00 | 32.22 | 56.25 | 30.48 |
| Time-MQA (Llama3-8B) | 53.05 | 64.15 | 50.77 | 32.56 |
| Time-MQA (Mistral-7B) | 53.66 | 69.81 | 49.37 | 24.62 |
| Time-R1 (Zhou et al., 2026) | 51.73 | 54.55 | 53.14 | 40.35 |
| Time-R1 (Liu et al., 2025d) | 50.53 | 55.00 | 66.48 | 30.10 |
| TimeOmni-1 | 67.66 | 69.44 | 63.07 | 70.48 |
| *Base: Qwen2.5-3B-Instruct* | | | | |
| Base | 40.99 | 26.67 | 61.36 | 31.43 |
| ChatTS (SFT) | 78.31 | **91.67** | 78.98 | 54.28 |
| **VeriTime (SFT+RL)** | 82.86 | 90.56 | 83.52 | 68.57 |
| Improvement (vs. Base) | +102.15% | +239.56% | +36.11% | +118.17% |
| *Base: Qwen3-4B-Instruct* | | | | |
| Base | 75.48 | 77.78 | 80.68 | 62.85 |
| ChatTS (SFT) | 82.21 | 89.44 | 80.68 | 72.38 |
| **VeriTime (SFT+RL)** | **86.55** | 91.11 | **87.50** | 77.14 |
| Improvement (vs. Base) | +14.67% | +17.14% | +8.45% | +22.74% |

## 4.2. Main Results (RQ1)

The performance of VeriTime compared to baselines on scenario-based tasks is presented in Table 2. The results show that general LLMs, including Qwen2.5 and Qwen3 in their base settings, perform equally unsatisfactorily, suggesting that general-purpose LLMs lack sufficient capacity for time series reasoning. Remarkably, VeriTime achieves performance competitive with frontier models such as GPT-5 and Gemini-3-Pro, despite being built on a significantly smaller backbone. The time series language models, Time-MQA (Kong et al., 2025) Time-R1 (Zhou et al., 2026), and TimeOmni (Guan et al., 2026) also yield suboptimal results, highlighting the necessity of integrating structured reasoning paths. With the VeriTime framework, Qwen2.5 achieves substantial improvements over the base model, with an overall gain of 102.15%, including 239.56% in anomaly detection and 118.17% in inferential calculation. Notably, Qwen3, which demonstrates stronger general reasoning abilities than Qwen2.5, consistently outperforms its original baseline across all tasks, achieving an overall gain of 14.67% and a 22.74% increase in inferential calculation.

Furthermore, we compare the performance of Qwen models fine-tuned on the original Q&A pairs from ChatTS, as shown in Table 2. Specifically, we adopt the same question set used to construct TSRBench, along with the corresponding original answers for supervised fine-tuning (SFT). The results indicate that while LLM finetuned by ChatTS achieves similar performance compared to that by TSR-Bench on the anomaly detection task, it underperforms on more complex tasks such as scenario attribution and inferential calculation. For Qwen3, the base and fine-tuned versions by the ChatTS dataset on the scenario attribution task perform nearly identically, revealing the limited utility of concise reasoning pairs for enhancing time series reasoning.

*Table 3.* Comparison of Accuracy (%) across four knowledge-based reasoning tasks.

| | CTU | ECG | EMG | RCW |
|---|---|---|---|---|
| *Classical Models* | | | | |
| Autoformer | 67.20 | 23.95 | 50.98 | 58.92 |
| FEDformer | 51.60 | 26.40 | 63.73 | 62.28 |
| PatchTST | 64.00 | 28.39 | 64.71 | 52.14 |
| iTransformer | 46.40 | 25.78 | 60.78 | 55.71 |
| TimesNet | 64.00 | 28.13 | **73.33** | 59.33 |
| DLinear | 52.40 | 26.82 | 47.06 | 39.55 |
| *General LLMs* | | | | |
| GPT-5 | 50.00 | 25.00 | 58.33 | 53.64 |
| Gemini-3-Pro | 50.85 | 28.39 | 61.64 | **65.31** |
| DeepSeek-R1-Distill-Qwen-7B | 34.75 | 21.58 | 33.58 | 28.88 |
| Qwen2.5-7B-Instruct | 50.85 | 16.84 | 43.80 | 29.95 |
| Meta-Llama3-8B-Instruct | 37.29 | 22.11 | 33.58 | 42.78 |
| Mistral-7B-v0.3 | 54.24 | 24.74 | 26.28 | 41.71 |
| GPT-4o-mini | 53.39 | 23.68 | 40.88 | 31.55 |
| *Time Series Language Models* | | | | |
| Time-MQA (Qwen2.5-7B) | 38.40 | 25.00 | 18.94 | 36.84 |
| Time-MQA (Llama3-8B) | 30.77 | 21.72 | 38.24 | 38.57 |
| Time-MQA (Mistral-7B) | 37.50 | 11.62 | 28.43 | 44.62 |
| Time-R1 (Zhou et al., 2026) | 59.17 | 23.74 | 42.86 | 33.52 |
| Time-R1 (Liu et al., 2025d) | 43.40 | 17.17 | 34.72 | 32.61 |
| TimeOmni-1 | 50.00 | 26.26 | 34.25 | 55.10 |
| *Base: Qwen2.5-3B-Instruct* | | | | |
| Base | 52.54 | 23.16 | 51.09 | 43.32 |
| **VeriTime (SFT+RL)** | 64.93 | 25.79 | 64.96 | 64.89 |
| Improvement | +23.58% | +11.36% | +27.15% | +49.79% |
| *Base: Qwen3-4B-Instruct* | | | | |
| Base | 42.50 | 15.15 | 56.46 | 43.08 |
| **VeriTime (SFT+RL)** | **67.50** | **30.30** | 65.31 | 63.59 |
| Improvement | +58.62% | +100.00% | +15.67% | +47.61% |

Besides, Table 3 presents the performance of VeriTime compared to baseline methods on knowledge-based reasoning tasks, which are inherently more challenging due to the need for real-world domain-specific knowledge to analyze category-wise differences in time series. VeriTime enables models to achieve significant improvements over their base performance, demonstrating that high-quality Chain-of-Thought reasoning paths can enhance time series reasoning. Additionally, VeriTime achieves performance that is superior to or comparable to traditional time series models. Notably, classical models lack generalization capabilities and require task-specific training and evaluation, whereas VeriTime can handle multiple tasks while providing detailed and interpretable reasoning paths for its predictions.

*Table 4.* Comparison of average token count per task during inference between the base model and VeriTime.

| | Anomaly Detection | Scenario Attribution | Inferential Calculation | TimeSeries Exam | CTU | ECG | EMG | RCW |
|---|---|---|---|---|---|---|---|---|
| Qwen3-4b-Instruct | 2192.46 | 1628.64 | 3782.56 | 3892.71 | 1691.25 | 7093.92 | 4584.15 | 3397.33 |
| **+VeriTime** | 632.67 | 619.88 | 620.30 | 695.52 | 1055.05 | 964.68 | 1004.71 | 990.29 |
| Token Reduction (%) | -71.14% | -61.94% | -83.60% | -82.13% | -37.62% | -86.40% | -78.08% | -70.85% |

*Table 5.* Comparison of Accuracy (%) on other benchmarks: Time-SeriesExam for synthetic time series reasoning and DROP for general numerical reasoning.

| | TimeSeriesExam | | DROP | |
|---|---|---|---|---|
| LLM | Qwen3 | Qwen2.5 | Qwen3 | Qwen2.5 |
| Base | 37.98 | 35.66 | 73.83 | 54.00 |
| SFT-only | 40.31 | 37.61 | 75.00 | 66.44 |
| **VeriTime** | 47.27 | 41.67 | 80.00 | 82.55 |
| Improvement (vs. Base) | +24.46% | +16.85% | +8.36% | +52.87% |

### 4.3. Analysis of TS-tailored Chain-of-Thought (RQ2)

To validate the TS-tailored Chain-of-Thought design, we first evaluate the performance of VeriTime on TimeSerie-sExam (Cai et al., 2024), a synthetic benchmark covering time series understanding tasks such as pattern recognition and causality analysis. As shown in Table 5, Qwen3 and Qwen2 achieve improvements of 24.46% and 16.85%, respectively. While SFT-only brings moderate gains, VeriiTime further improves performance by a clear margin, suggesting that GRPO contributes meaningfully beyond SFT alone and equips LLMs with stronger ability to reason about TS patterns across tasks. We also assess VeriTime on DROP (Dua et al., 2019) for numerical reasoning, where it notably improves the performance of Qwen2.5 from 54.00% to 82.55%, thereby highlighting VeriTime's effectiveness in extending LLMs' analytical capabilities to the TS domain.

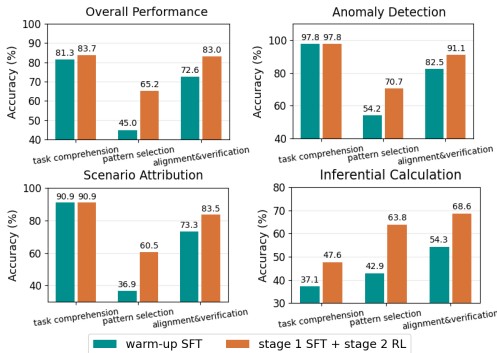

*Figure 3.* Step-wise Accuracy (%) comparison of Qwen2.5-3B-Instruct: warm-up SFT vs. stage 1 SFT + stage 2 RL on scenario-based reasoning tasks.

We further evaluate VeriTime's reasoning quality via the step-level correctness metric as illustrated in Figure 3. Specifically, we compare the step-wise accuracy of Qwen2.5 trained with VeriTime on critical steps, including task comprehension, pattern selection, answer alignment, and verification, to that of Qwen2.5 fine-tuned with 10% of the data for warm-up training. The warm-up LLM achieves ac-

curacy comparable to VeirTime in the task comprehension step for anomaly detection and scenario attribution tasks. However, a significant gap remains in pattern selection accuracy (65.2% vs. 45.0%), suggesting that the warm-up LLM still struggles in identifying critical patterns. This results in lower performance in answer alignment and verification, with the warm-up model achieving only 54.3% accuracy compared to 68.6% for VeriTime in the inferential calculation. These results demonstrate that recognizing task-relevant patterns is fundamental for final results and supports deeper analysis of time series segments.

The TS-tailored Chain-of-Thought improves both task performance and inference efficiency by substantially reducing token usage. As shown in Table 4, the base Qwen3 model produces excessively long reasoning processes when handling time series tasks, and often generates lengthy and unfocused analyses of time series segments. More challenging tasks tend to induce even longer reasoning paths in the base model without corresponding improvements in final inference accuracy. For instance, average token counts reach 3,782 for inferential calculation and 7,093 for ECG tasks. In contrast, the proposed TS-tailored CoT guides LLMs in a more structured manner, achieving an average token reduction rate of 71% across 8 tasks. This demonstrates that VeriTime enhances reasoning performance while significantly improving efficiency.

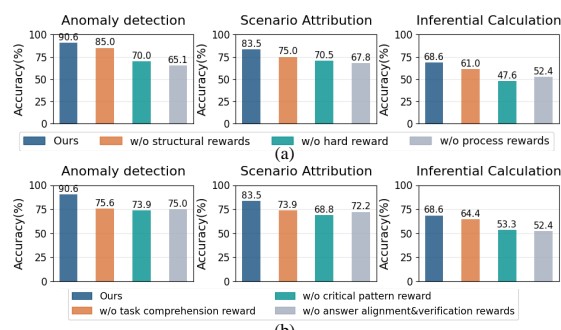

*Figure 4.* Ablation results on Qwen2.5-3B-Instruct. (a) Performance without structural, hard, or process rewards. (b) Performance without task comprehension, critical pattern, or alignment& verification rewards.

### 4.4. Studies of Reward Composition (RQ3)

To evaluate the effectiveness of VeriTime's multi-objective reward design, we sequentially ablate the structural, hard, and process rewards to analyze their individual contributions. As illustrated in Figure 4 (a), the exclusion of structural rewards yields the smallest performance degradation, pri-

marily resulting in unstructured or overly lengthy responses. In contrast, removing hard or process rewards causes more pronounced declines, with the absence of hard rewards dropping inferential calculation accuracy from 68.6% to 47.6%. These results highlight the crucial role of multi-objective design in enhancing reasoning capabilities during RL training, as it directly influences the correctness of the final answer.

We further analyze individual components within the process rewards in Figure 4 (b). Removing the critical pattern reward results in an 18.94% average drop across all tasks, indicating that identifying task-specific patterns is essential for answer correctness and subsequent analysis of time series segments. Excluding the task comprehension reward degrades anomaly detection and scenario attribution more than inferential calculation, likely because task understanding is more crucial in the former two tasks. For instance, anomaly detection requires distinguishing between normal or abnormal states, while scenario attribution relies on accurately interpreting the contextual conditions. Finally, removing answer alignment and verification rewards drops inferential calculation performance by 23.62%, indicating that quantitative reasoning relies more heavily on supervision signals aligned with correct numerical labels.

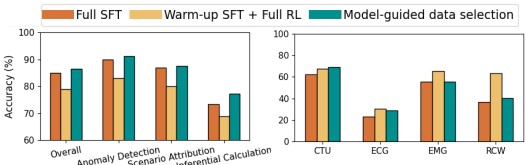

*Figure 5.* Performance comparison of training data selection methods on scenario-based reasoning tasks and knowledge-based reasoning tasks for the Qwen3-4B-Instruct model.

### 4.5. Analysis of Data Scheduling (RQ4)

We analyze how data scheduling strategy influences the trade-off between performance and efficiency by comparing two strategies based on the sample difficulty hierarchy of TSRBench: (1) model-guided data selection for scenario-based reasoning, where a warm-up model routes failed cases to RL and successful ones to SFT; and (2) full RL for knowledge-based reasoning, combining warm-up SFT with RL throughout. We evaluate both strategies across task types, with overall results in Figure 5 and stage-wise contributions detailed in Figure 6 and Figure 7.

For scenario-based tasks, model-guided data scheduling consistently outperforms naive SFT on the full dataset, whereas full RL leads to performance degradation. This indicates that extensive RL on medium-difficulty cases may induce a less stable learning process. These tasks benefit more from general reasoning pathways via supervised training, complemented by a few hard cases for reasoning enhancement. In the model-guided strategy, RL cases are only **0.25×** of SFT cases. As shown in Figure 6, the performance improvements for each task are predominantly driven by the SFT stage for

Qwen2.5, with increases of 58.9% for anomaly detection and 29.5% for inferential calculation.

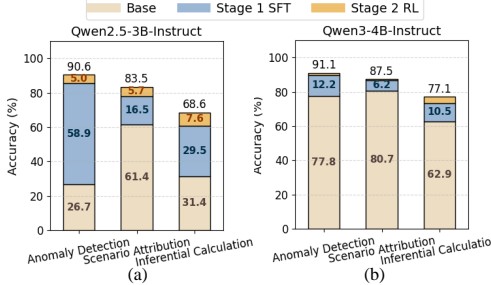

*Figure 6.* Performance breakdown on scenario-based tasks, illustrating stage contributions to overall accuracy (%) for the model-guided data selection method for (a) Qwen2.5 and (b) Qwen3.

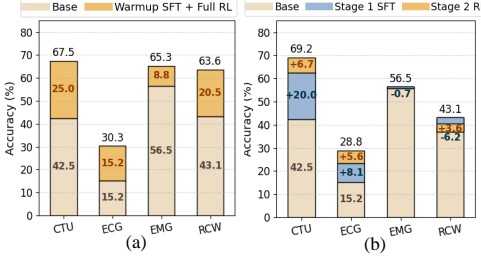

*Figure 7.* Performance breakdown on knowledge-based reasoning tasks for Qwen3-4B-Instruct, illustrating stage contributions for (a) full RL and (b) model-guided data selection.

For knowledge-based tasks, both full RL and model-guided selection outperform full SFT, with full RL achieving over 20% improvement on the RCW subset (Figure 7 (a)). Under model-guided allocation, RL receives **1.4×** as many cases as SFT. Conversely, SFT alone causes performance drops of 0.7% and 6.2% on EMG and RCW (Figure 7 (b)). This indicates that tasks that require subtle pattern discrimination over longer time segments and domain expertise may not be adequately addressed by supervised training alone. Sufficient RL exposure provides the model with a broader exploration space, enabling it to better identify and reason over task-critical time series patterns. These findings highlight the importance of adaptive SFT-RL data allocation for time series reasoning. Besides, we analyze the performance stability of VeriTime in Appendix A.

## 5. Conclusion

In this paper, we propose VeriTime, a novel framework that enhances time series reasoning capabilities of LLMs through data synthesis, scheduling, and multi-objective RL training. We design a data synthesis pipeline to construct a multi-modal dataset with process-verifiable annotations, followed by a two-stage RFT strategy incorporating multi-objective rewards and selective rollout to balance performance and efficiency. Extensive experiments demonstrate that VeriTime enables moderate-sized LLMs to achieve substantial performance improvements, establishing a promising direction for leveraging LLM reasoning in time series analysis.

## Acknowledgements

We would like to thank anonymous reviewers and area chairs for their constructive comments and efforts in improving our paper. We also gratefully acknowledge the computational resources and support provided by the National Supercomputer Center in Guangzhou.

## Impact Statement

This paper aims to advance the reasoning capabilities of LLMs in the time series domain. There are many potential societal consequences of our work, none of which we feel must be specifically highlighted here.

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

# Appendix

This appendix contains: 1) Section A: additional experiment results and analysis; 2) Section B: details of the TSRBench reasoning dataset; 3) Section C: implementation details of VeriTime; 4) Section D: limitations; and 5) Section E: prompt templates for TSRBench curation. The code for VeriTime is available at `https://github.com/ZhoujhZoe/VeriTime`.

# A. Additional Experimental Results

In the section, we present extended analyses of VeriTime. Subsection A.1 examines the performace stability of VeriTime across parameter sensitivity, prompt sensitivity, and generation settings. Subsection A.2 analyzes the data selection strategy. Subsection A.3 analyzes the architecture-agnostic effectiveness of VeriTime when applied to time series language models. Subsection A.4 evaluates the zero-shot reasoning performance of VeriTime on real-world time series data. Subsection A.5 illustrates the training dynamics of VeriTime.

## A.1. Performance Stability Analysis of VeriTime

**Inference Parameter Sensitivity.** To assess the stability of VeriTime across different inference configurations, we evaluate the model's performance under varying temperature settings (0.2, 0.4, 0.6, and 0.8). As reported in Table 6, standard deviations remain negligible (0.00% to 0.78%) on scenario-based tasks. For knowledge-based tasks, while the standard deviations are slightly higher, with the ECG task reaching 2.05%, they remain within acceptable ranges. The results indicate that the performance of VeriTime remains stable despite variations in temperature settings.

*Table 6.* Average performance and standard deviation of Qwen3-4B-Instruct model inference over different temperature settings.

| | Scenario-based | | | Knowledge-based | | | |
|---|---|---|---|---|---|---|---|
| | Anomaly Detection | Scenario Attribution | Inferential Calculation | CTU | ECG | EMG | RCW |
| **Average Accuracy (%)** | 91.11 | 87.22 | 76.19 | 67.34 | 31.91 | 64.91 | 63.59 |
| **Standard Deviation (%)** | 0.00 | 0.57 | 0.78 | 0.98 | 2.05 | 1.13 | 0.00 |

**Prompt Sensitivity.** We investigate the performance of VeriTime to prompt variations through ablation studies: (1) paraphrasing the system prompt while preserving its semantic meaning, and (2) introducing minor modifications to the question (e.g., removing the *you are a time series expert* statement).

The results presented in Table 7 indicate that VeriTime exhibits robustness to system prompt paraphrasing. While the paraphrased system prompt exhibits greater sensitivity relative to paraphrased task instructions on knowledge-based tasks, the model achieves 71.13% accuracy on the EMG task. These observations show that VeriTime does not exhibit strong dependence on phrasing conventions, indicating its robustness to prompt variations across different tasks.

*Table 7.* Performance evaluation of VeriTime across prompt variations with task-wise Accuracy (%) on Qwen3-4B-Instruct.

| | Scenario-based | | | Knowledge-based | | | |
|---|---|---|---|---|---|---|---|
| | Anomaly Detection | Scenario Attribution | Inferential Calculation | CTU | ECG | EMG | RCW |
| **Original Prompt** | **91.11** | **87.50** | 77.14 | **67.50** | **30.30** | 65.31 | **63.59** |
| **Paraphrased System Prompt** | 87.15 | 84.09 | 74.29 | 59.63 | 24.74 | **71.13** | 56.92 |
| **Paraphrased Question** | 89.39 | 85.23 | **80.77** | 60.00 | 27.41 | 61.70 | 58.85 |

**Generation Setting.** For parameter analysis, we further conduct an ablation study on the number of generations $G$ per input during GRPO training. Table 8 shows how varying $G$ affects performance on scenario-based tasks. Performance generally increases with larger $G$, as more generations yield more stable and representative reward estimates across different reasoning trajectories. A slight drop emerges when $G = 5$, likely due to overthinking on medium-difficulty tasks, which aligns with our findings in RQ4. To balance performance and computational cost, we set $G = 4$ as the default in all main experiments.

*Table 8.* Effect of generation number $G$ on performance (Accuracy %) across scenario-based tasks.

| Generation Number | Overall | Anomaly Detection | Scenario Attribution | Inferential Calculation |
|---|---|---|---|---|
| 2 | 85.47 | 90.00 | 86.93 | 75.24 |
| 3 | 86.33 | 93.33 | 84.66 | 77.14 |
| 4 | 86.55 | 91.11 | 87.50 | 77.14 |
| 5 | 83.30 | 87.78 | 85.80 | 71.43 |

*Table 9.* Performance Comparison of different data selection strategies on scenario-based tasks for Qwen3-4B-Instruct.

| | Overall | Anomaly Detection | Scenario Attribution | Inferential Calculation |
|---|---|---|---|---|
| Random | 83.30 | 86.67 | 84.09 | 76.19 |
| Token Length | 85.25 | 88.89 | 86.93 | 76.19 |
| Perplexity | 84.82 | 90.56 | 83.52 | 77.14 |
| IFD | 84.16 | 88.33 | 81.82 | **80.95** |
| Model-guided | **86.55** | **91.11** | **87.50** | 77.14 |

## A.2. Analysis of Data Selection Strategy

Furthermore, we compare the model-guided data selection strategy with several baselines on scenario-based tasks: (1) Random Sampling; (2) TokenLength (Xia et al., 2025), which ranks samples by token length; (3) PPL, which selects samples with the highest perplexity under a pretrained model; and (4) Instruction-Following Difficulty (IFD) (Li et al., 2024), which measures the difficulty of instructional samples for the model. For a fair comparison, each method selects the same amount of SFT data as the model-guided strategy, with the remaining samples allocated to RL. Results are provided in Table 9, where the random sampling consistently underperforms. The model-guided strategy shows robust performance and generalization across tasks, reliably identifying high-quality RL data to enhance LLM time series reasoning with minimal training overhead while maintaining efficiency.

## A.3. Architecture-Agnostic Effectiveness of VeriTime

To isolate the gains attributable to VeriTime from those arising from differences in model architectures and training corpora, we apply VeriTime to two representative time series language models, ITFormer (Wang et al., 2025b) and OpenTSLM (Langer et al., 2026), and compare it against their original training

*Table 10.* Performance comparison of VeriTime and baseline training algorithms across two TSLM backbones on TimeSeriesExam (Accuracy %).

| | OpenTSLM | ITFormer-0.5B | ITFormer-3B |
|---|---|---|---|
| Base training protocol | 28.12 | 33.33 | 36.43 |
| + VeriTime | 46.22 | 44.96 | 44.03 |

algorithms under a shared training and evaluation protocol. All methods are trained on ChatTS (Xie et al., 2025) and evaluated on the held-out TimeSeriesExam (Cai et al., 2024), ensuring that all reported results reflect performance on unseen data. Each baseline follows its original training protocol, while VeriTime freezes the time series encoder and trains only the LLM component through SFT on process-verifiable CoT trajectories, followed by a process-verifiable RL stage. This design isolates the reasoning supervision to the language modeling component, attributing performance differences to the training mechanism rather than to encoder capacity. As shown in Table 10, replacing the baseline training algorithm with VeriTime yields consistent improvements across all configurations. The consistent gains across heterogeneous architectures and parameter scales indicates that process-verifiable reward supervision provides a reliable and architecture-agnostic source of improvement for time series reasoning, complementary to advances in TSLM architecture design.

## A.4. Case Study of VeriTime on Real-World Time Series

To evaluate the **zero-shot** reasoning capabilities of LLMs on real-world time series data and examine their practicality for real-world deployment, we conduct case studies on the Qwen3-4B-Instruct model trained under VeriTime on TSRBench. Specifically, we utilize the Time-MMD (Liu et al., 2024a) dataset, which integrates multiple rigorously curated and verified data sources up to May 2024, to perform an in-depth assessment of time series performance in real-world applications.

For anomaly detection tasks, we assess the model's ability to identify abnormal patterns in TS from both holistic and threshold-based perspectives. As illustrated in Figure 8, the model first analyzes a financial series where a normal trend is defined as *free of sharp, atypical declines*. It successfully pinpoints two significant drops within the overall upward

trend, accurately locating the sharpest decline and quantifying its magnitude. This demonstrates the model's fine-grained understanding of temporal dynamics and its effectiveness in zero-shot anomaly detection. When applied to the PM2.5 AQI index under a threshold-based setting, the model further detects value spikes, reports corresponding timestamps and magnitudes, and reasons that sustained high AQI levels pose greater health risks than isolated peaks. These results collectively highlight the model's potential for real-world anomaly detection applications.

For inferential calculation tasks, the model is first prompted to analyze the 532-month retail broiler composite from 1980 to 2024 and identify distinct upward trends over the entire period. As illustrated in Figure 9, it successfully recognizes three major trends based on directional movement, duration, and amplitude, and accurately estimates their approximate start and end times. When applied to a traffic volume dataset with clear annual cycles, the model further segments the data by year, detects primary and secondary peaks within each cycle, and correctly determines the month in which traffic volume typically reaches its maximum. These results demonstrate the model's capability to reason over long temporal horizons and capture complex periodic patterns.

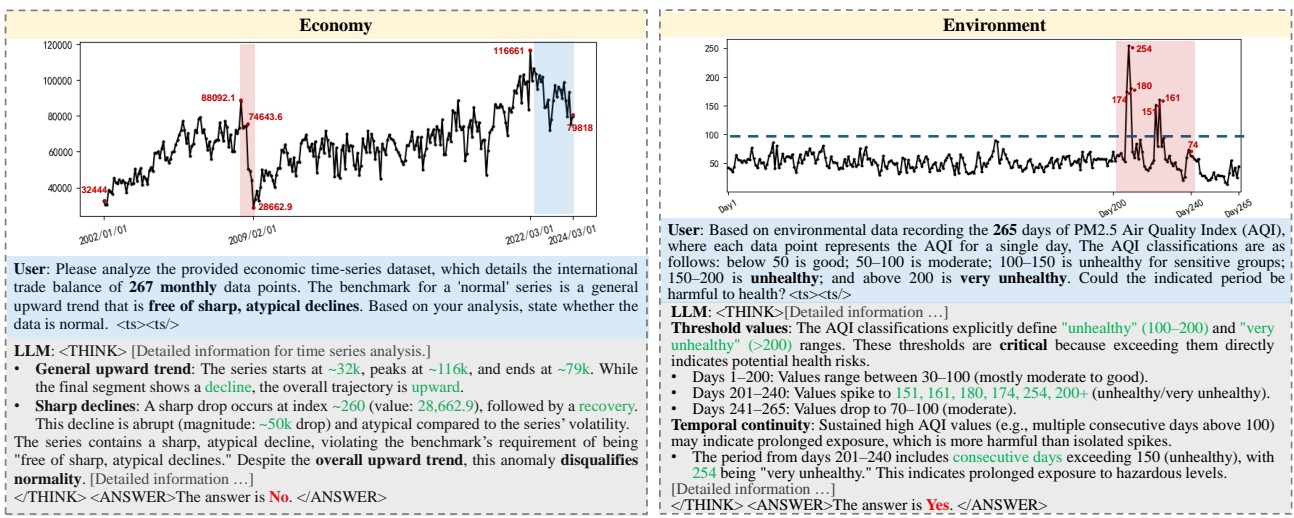

*Figure 8.* Case studies on anomaly detection tasks in economy and environment domains.

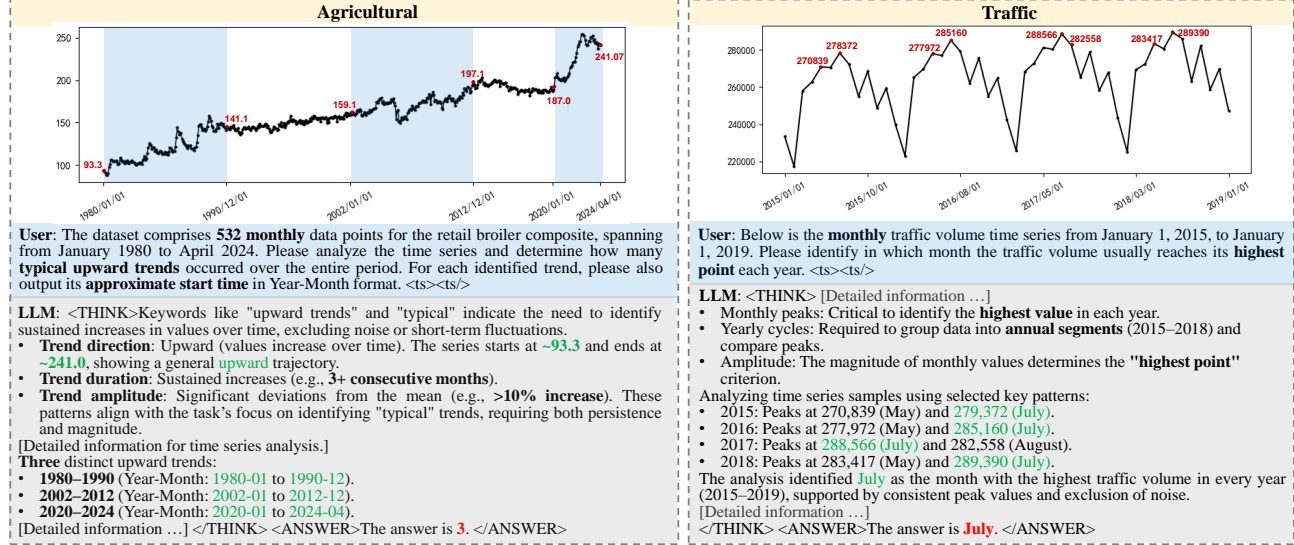

*Figure 9.* Case studies on inferential calculation tasks in agricultural and traffic domains.

### A.5. VeriTime Training Dynamics

To evaluate the training dynamics of VeriTime, Figure 10 presents the accuracy of Qwen3-4B-Instruct during RL training across four knowledge-based reasoning tasks. To reduce evaluation cost during RL, we report performance on a subset of the test set, which may exhibit slight variance compared to the full experiment results. The curves reveal initial performance fluctuations in the early training phase, followed by consistent improvement as training progresses. Overall, all subsets demonstrate a clear view of accuracy gains, indicating stable and effective optimization.

Besides, Figure 11 presents the reinforcement learning training dynamics of VeriTime, illustrating four key metrics: length reward, final answer reward, process answer reward, and total reward. The black trend lines indicate consistent optimization as training progresses across all reward components, showing the performance of VeriTime remains stable and improve gradually with training steps increase.

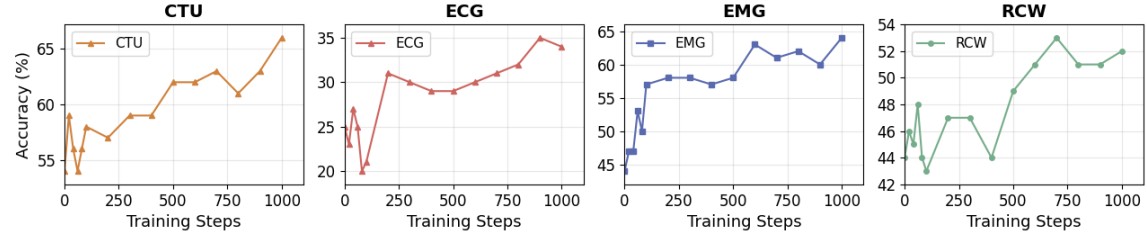

*Figure 10.* Accuracy (%) over training steps on knowledge-based reasoning tasks.

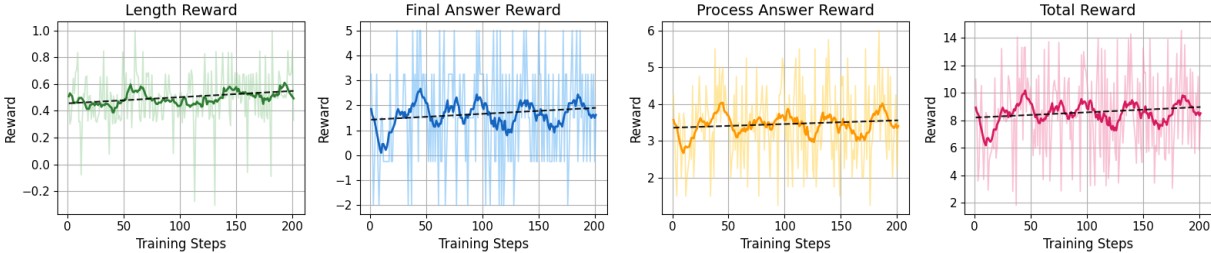

*Figure 11.* Reinforcement Learning training dynamics for VeriTime.

## B. Details of TSRBench

### B.1. Process Label Distribution

TSRBench provides diverse intermediate reasoning signals, comprising **9,746** occurrences across **2,684** unique process pattern labels. The top ten most frequent patterns are detailed in Table 11. To ensure data quality, we rigorously filter all samples for final-answer correctness and manually audited approximately **45%** of the process labels to remove invalid annotations. All examples in TSRBench are equipped with complete step-by-step reasoning labels.

*Table 11.* Top-10 most frequent process pattern labels in TSRBench.

| Pattern | Count | Rate (%) | Pattern | Count | Rate (%) |
|---------|-------|----------|---------|-------|----------|
| Amplitude | 563 | 5.78 | Threshold Value | 211 | 2.16 |
| Fluctuation | 486 | 4.99 | Downward Spike | 207 | 2.12 |
| Continuity | 460 | 4.72 | Anomaly | 133 | 1.36 |
| Trend | 377 | 3.87 | Steady Trend | 94 | 0.96 |
| Upward Spike | 237 | 2.43 | Deviation | 82 | 0.84 |

### B.2. Cross-Model Validation

To validate the quality of TSRBench synthesized by our data generation pipeline TSRgen, we conduct a cross-model evaluation of the intermediate reasoning steps produced by the expert model (DeepSeek-R1). Specifically, we use three

state-of-the-art LLMs: Qwen3-235B, GPT-5, and Gemini-2.5, which serve as independent judges to assess the reasoning quality of the teacher model's Chain-of-Thought trajectories. The full template is provide in subsection E.2.

**Evaluation Protocol.** For each subtask in TSRBench, we randomly sample 100 instances for quality evaluation and subsequently compute the average score for each LLM. Each judge model assesses reasoning quality across five dimensions. The first two dimensions focus on verifying the correctness of process-verifiable labels extracted from the expert model, while the latter three evaluate the quality of the reasoning process itself:

1. **Task Intent Alignment**: Whether Step 1 correctly identifies the task objective and required outcome.

2. **Key Pattern Relevance**: Whether Step 2 captures task-relevant temporal cues (e.g., thresholds, trends, patterns).

3. **Logical Coherence**: Whether reasoning transitions and justifications in Steps 3–6 maintain consistency without logical gaps or contradictions.

4. **Robustness vs. Shortcuts**: Whether the reasoning avoids spurious shortcuts or lucky guesses and performs necessary verification checks.

5. **Bias/Artifact Check**: Whether the reasoning contains unwarranted assumptions or biases unrelated to the data or task.

The **scoring rubric** is defined as follows: 5 — fully correct, precise, and well-justified with no hallucinations; 4 — overall correct and well-grounded, with only minor omissions; 3 — mixed quality, containing unsupported claims or missing checks; 2 — major issues, including key omissions or likely use of shortcuts; 1 — largely incorrect or hallucinated.

**Results.** Table 12 presents the cross-model evaluation results. The scores indicate a strong inter-judge agreement and consistently high quality across all dimensions. Notably, all three judge models assign high scores ($> 4.5$) for *Intent Alignment* and *Pattern Relevance* in both reasoning task categories, validating the correctness of the process-verifiable labels extracted from DeepSeek-R1's reasoning trajectories. The remaining three dimensions, which assess the overall quality of TS-tailored Chain-of-Thought in terms of logical coherence and robustness, also achieve consistently high scores, with only GPT-5 showing slightly lower scores on knowledge-based tasks. These results provide strong evidence for the reliability and quality of the intermediate reasoning paths of TSRBench.

*Table 12.* Cross-model evaluation of DeepSeek-R1's reasoning quality on TSRBench.

| LLM | Evaluation Metrics | | | | |
|---|---|---|---|---|---|
| | Intent Alignment | Pattern Relevance | Logical Coherence | Robustness | Bias Check |
| *Scenario-based Task* | | | | | |
| Qwen3-235B | 4.99 | 4.96 | 4.70 | 4.56 | 4.70 |
| GPT-5 | 4.97 | 4.85 | 4.50 | 4.30 | 4.62 |
| Gemini-2.5 | 5.00 | 4.98 | 4.31 | 4.38 | 4.62 |
| *Knowledge-based Task* | | | | | |
| Qwen3-235B | 5.00 | 4.99 | 4.21 | 4.18 | 4.05 |
| GPT-5 | 4.94 | 4.58 | 3.92 | 3.68 | 4.03 |
| Gemini-2.5 | 5.00 | 4.99 | 4.07 | 4.14 | 4.99 |

## B.3. Dataset Construction and Illustrative Examples

The objective of TSRBench is to curate process-verifiable TS–text reasoning tasks that encompass diverse domains. To achieve this, we design two categories of reasoning tasks: **scenario-based** and **knowledge-based** tasks, derived from synthetic datasets and real-world datasets, respectively.

For scenario-based tasks, previous work like ChatTS (Xie et al., 2025) primarily focuses on training LLMs to understand TS patterns rather than to address more complex reasoning tasks, resulting in non-trivial variance in difficulty and annotation quality. Building on this, we filter and reorganize the original data to construct three distinct types of of scenario-based reasoning tasks: *anomaly detection*, *scenario attribution*, and *inferential calculation*. For knowledge-based tasks, we aim to evaluate the ability of LLMs to analyze real-world, domain-specific time series. To this end, we curate four representative datasets (*CTU, ECG, EMG, and RCW*) from various domains and standardize the raw data into Q&A classification tasks. Examples of the generated reasoning questions are provided in Table 13.

*Table 13.* Illustrative examples of the reasoning tasks in TSRBench.

| Task | Examples |
| --- | --- |
| Deductive Reasoning Anomaly Detection | You are a time series analysis expert. This is a metric called Cloud Cover collected from **Weather Forecasting** with length of 256: <ts><ts/>. If **a sudden and significant increase** in cloud cover is considered an anomaly, should the behavior at **point 204** to **point 206** be treated as an anomaly? |
| Causal Reasoning Scenario Attribution | You are a time series analysis expert. This is a metric called Carbon Emissions collected from **Energy** with length of 256: <ts><ts/>. Considering the temporal correlation in the Carbon Emissions time series, which of the following could be a **potential cause** for the observed rapid rise in emissions followed by a slow decline starting around point 219? **Choose from**: A) Introduction of new industrial activities, B) Implementation of strict environmental regulations, C) Seasonal decrease in industrial activities. |
| Quantitative Reasoning Inferential Calculation | You are a time series analysis expert. This is a metric called Parallel Query Performance collected from **Oracle Database** with length of 256: <ts><ts/>. In a database system, the **Parallel Query Performance** is monitored to ensure optimal operation. The performance data starts from June 1, and each point represents a minute. During normal operation, the performance is expected to **remain steady**. However, there are rare instances where the system experiences **a sudden drop** in performance. **How many** such critical performance drops, defined as a rapid decline from a peak, are observed in the time series? |
| Inductive Reasoning Classification | You are analyzing an **audio signal** to determine the presence of right whale vocalizations. Up-calls are the most commonly documented right whale vocalisation with an acoustic signature of approximately 60Hz-250Hz, typically lasting 1 second. **Right whale calls** can often be difficult to hear as the low-frequency band can become congested with anthropogenic sounds such as ship noise, drilling, piling, or naval operations. The signal shows a two-second waveform sampled at 2kHz, resulting in a time series of length 500: <ts><ts/>. Your goal is to **classify** whether the waveform segment contains a right whale call. Based on the analysis, choose the best matching label for the full signal from: A) Right Whale Present, B) No Right Whale. |

Overall, TSRBench evaluates LLMs across multiple reasoning abilities, such as deductive reasoning involves inferring conclusions from predefined conditions, causal reasoning identifies the most plausible cause based on the observed series, quantitative reasoning requires accurate numerical analysis, and inductive reasoning compares time series patterns to predict categories in real-world data.

## C. Implementation Details of VeriTime

### C.1. More Details of Multi-Objective Reward Design

VeriTime adopts a set of fine-grained reward components designed for multi-objective optimization, categorized into structural rewards $r_{\text{struct}}$, the hard reward $r_{\text{hard}}$, and process rewards $r_{\text{process}}$. The overview of the multi-objective reward design is provided in Table 14.

**Format Reward.** $r_{\text{fmt}}$ enforces adherence to the required output template by verifying that the reasoning path is within <THINK> tags and the final answer in <ANSWER> tags. Fully compliant outputs receive a positive reward, while structural violations are penalized, with stronger incentives for valid structures over discouraging violations.

$$r_{\text{fmt}}(y) = \begin{cases} +3.0, & y \in \mathcal{S}_{\text{valid}}, \\ -2.0, & \text{otherwise.} \end{cases} \tag{11}$$

**Length Reward.** $r_{\text{len}}$ encourages responses of appropriate detail and depth, penalizing outputs that are too short or overly long with repetitive or meaningless content. Let $L(y)$ denote the token length of output $y$, with penalties for deviations from the desired range defined by $L_{\text{thr}}$ (minimum length) and $L_{\text{tol}}$ (upper tolerance threshold):

$$r_{\text{len}}(y) = \begin{cases} -2.0, & L(y) \geq L_{\text{tol}}, \\ +1.0, & L_{\text{thr}} \leq L(y) < L_{\text{tol}}, \\ \frac{L(y)}{L_{\text{tol}}}, & L(y) < L_{\text{thr}}. \end{cases} \tag{12}$$

The values are set to balance positive reinforcement for desirable lengths with penalties for violations. The structural rewards

*Table 14.* Overview of the reward components in the VeriTime RL training framework.

| Category | Reward Component | Symbol | Description |
|---|---|---|---|
| Structural Rewards | Format Reward | $r_{\text{fmt}}$ | Enforces strict adherence to the required `<THINK>` and `<ANSWER>` structure. |
| | Length Reward | $r_{\text{len}}$ | Encourages responses within a desirable length range, discouraging both insufficient analysis and or redundant content. |
| Hard Reward | Hard Reward | $r_{\text{hard}}$ | Provides strong supervision on final-answer correctness via exact matching of the ground truth. |
| Process Rewards | Task Comprehension Reward | $r_{\text{intent}}$ | Measures whether the model correctly interprets the task objective and underlying requirements. |
| | Critical Pattern Reward | $r_{\text{pattern}}$ | Assesses identification of key reasoning patterns, assigning a higher reward for capturing more essential patterns. |
| | Answer Alignment Reward | $r_{\text{align}}$ | Rewards intermediate answers that correctly align with the ground truth. |
| | Answer Verification Reward | $r_{\text{verify}}$ | Rewards correctness of the final self-verified answer produced after self-reflection. |

$r_{\text{struct}}(y)$ is defined as:

$$r_{\text{struct}}(y) = r_{\text{fmt}}(y) + r_{\text{len}}(y). \tag{13}$$

**Hard Reward.** $r_{\text{hard}}$ evaluates the factual correctness of the predicted final answer $A(y)$ from the `<ANSWER>` tags. As the primary optimization criterion, it has the highest magnitude and provides strong supervision for end-task accuracy. Correct answers receive a high reward, while incorrect ones are moderately penalized to encourage exploration, allowing the model to retain exploratory behavior. Let $\hat{a}$ denote the ground truth:

$$r_{\text{hard}}(y) = \begin{cases} +5.0, & A(y) = \hat{a}, \\ -2.0, & \text{otherwise}, \end{cases} \tag{14}$$

Together with the process reward described in subsection 3.3, we aggregate the three reward components into the global reward signal $R(y)$ to ensure structural fidelity, answer correctness, and validity of intermediate reasoning steps. The aggregation uses a uniform average, as the relative importance of each component is reflected in their designed magnitudes. The GRPO training pipeline for VeriTime is detailed in Algorithm 1.

$$r(y) = r_{\text{struct}}(y) + r_{\text{hard}}(y) + r_{\text{process}}(y). \tag{15}$$

### C.2. Policy Optimization Using GRPO

VeriTime adopts the Group Relative Policy Optimization (GRPO) (Shao et al., 2024) algorithm to improve the stability and efficiency of the training process. Unlike PPO (Schulman et al., 2017) that relies on a separately trained value network, GRPO normalizes advantages across a group of rollouts, providing a more stable and self-contained optimization framework. Formally, let $\pi_\theta$ denote the curren model policy with parameters $\theta$, and $\pi_{\text{ref}}$ be a frozen reference policy (e.g., a pre-trained or SFT model). For each input context $x$, we sample a group of $G$ complete output sequences from the old policy $\{y_i\}_{i=1}^G \sim \pi_{\text{old}}(\cdot|\mathbf{x})$, and assign each generated sequence a scalar reward $r_i$ based on our composite reward function. The group-normalized advantage is computed as:

$$\hat{A}_i = \frac{r_i - \mu_r}{\sigma_r}, \quad \mu_r = \frac{1}{G}\sum_{j=1}^G r_j, \quad \sigma_r = \sqrt{\frac{1}{G}\sum_{j=1}^G (r_j - \mu_r)^2 + \varepsilon}. \tag{16}$$

where each token within the same sequence $y_i$ shares the same normalized advantage $\hat{A}_i$, ensuring stable gradient scaling across contexts.

To update the policy $\pi_\theta$ using this advantage, we employ a clipped counterpart, which helps prevent large, detrimental policy updates. The per-sample clipped objective is given by:

$$\mathcal{L}_{\text{clip}}(\theta) = \min\left(\rho_{i,k}\hat{A}_i, \, \text{clip}(\rho_{i,k}, 1-\epsilon, 1+\epsilon)\hat{A}_i\right), \tag{17}$$

---

**Algorithm 1** GRPO Training Pipeline of VeriTime

---

**Require:** Initial time-series LLM $\pi_\theta$, dataset $\mathcal{D}$, group size $G$, temperature $\tau$, learning rate $\eta$
1: Supervised fine-tune $\pi_\theta$ on structured time-series data with format tags $\langle\text{THINK}\rangle$, $\langle\text{ANSWER}\rangle$
2: **for** each RL iteration **do**
3:     Update the reference model: $\pi_{\text{ref}} \leftarrow \pi_\theta$
4:     **for** Step $= 1, 2, \ldots$ **do**
5:         Sample a mini-batch $\mathcal{B}$ from $\mathcal{D}$
6:         Update the old model: $\pi_{\text{old}} \leftarrow \pi_\theta$
7:         Sample $G$ outputs $\{\mathbf{y}_i\}_{i=1}^G \sim \pi_{\theta_{\text{old}}}(\cdot \mid \mathbf{X}, \mathbf{q})$ for each time-series instance $(\mathbf{X}, \mathbf{q}) \in \mathcal{B}$
8:         **for** each sampled $\mathbf{y}_i$ **do**
9:             Parse tags: $\langle\text{THINK}\rangle_i$, $\langle\text{ANSWER}\rangle_i = \hat{c}_i$
10:           Compute format reward: $r_i^{\text{fmt}} = \mathbb{I}[\text{tags well-formed and non-empty}]$
11:           Compute length reward: $r_i^{\text{len}} = \mathbb{I}[\text{threshold} < |\mathbf{y}_i| < \text{max\_len}]$
12:           Compute hard reward: $r_i^{\text{hard}} = \mathbb{I}[\hat{c}_i = c^*]$
13:           Compute process reward: $r_i^{\text{proc}} = \sum_{j=1}^{N_{\text{steps}}} \mathbb{I}[\text{step}_j\text{-judgment} \approx \text{label}_j]$
14:           Compute composite reward: $r_i = \lambda^{\text{fmt}} r_i^{\text{fmt}} + \lambda^{\text{len}} r_i^{\text{len}} + \lambda^{\text{hard}} r_i^{\text{hard}} + \lambda^{\text{proc}} r_i^{\text{proc}}$
15:         **end for**
16:         Compute $\{\hat{A}_i\}_{i=1}^G$ for each group via advantage estimation
17:         Update $\pi_\theta$ by maximizing $\mathcal{L}(\theta) = \frac{1}{G|\mathcal{B}|} \sum_{(\mathbf{X},\mathbf{q}) \in \mathcal{B}} \sum_{i=1}^G \hat{A}_i \log \pi_\theta(\mathbf{y}_i \mid \mathbf{X}, \mathbf{q})$
18:     **end for**
19: **end for**
20: **return** $\pi_\theta$

---

where $\rho_{i,k}$ is the importance sampling ratio for each token, and $\epsilon$ controls the clipping range:

$$\rho_{i,k} = \frac{\pi_\theta(y_{i,k} \mid y_{i,<k}, \mathbf{x})}{\pi_{\text{old}}(y_{i,k} \mid y_{i,<k}, \mathbf{x})}. \tag{18}$$

The overall objective function $\mathcal{L}_{\text{GRPO}}(\theta)$ is then maximized during training. This objective balances the expected clipped advantage with a KL-divergence penalty against the reference policy $\pi_{\text{ref}}$:

$$\mathcal{L}_{\text{GRPO}}(\theta) = \frac{1}{G} \sum_{i=1}^G \frac{1}{|y_i|} \sum_{k=1}^{|y_i|} \left[ \mathcal{L}_{\text{clip}}(\theta) - \beta \, \text{KL} \left[ \pi_\theta \,\|\, \pi_{\text{ref}} \right] \right], \tag{19}$$

where $\beta$ controls the strength of the KL penalty and ensures that the policy update remains close to the reference model. This objective guides the policy toward higher rewards, leveraging stable GRPO advantage estimates within a constrained optimization framework.

### C.3. Training Configuration

All training and inference for VeriTime and baseline models are performed on 2×A100 80G GPUs. The base LLM for VeriTime training is Qwen2.5-3B-Instruct, and Qwen3-4B-Instruct, which exhibit stronger general reasoning abilities than the Qwen2.5 series. The maximum sequence length for the LLMs is set to 10K tokens. Key hyperparameters of the overall training and inference process are summarized in Table 15.

## D. Limitations

In this study, we investigate the impact of data scheduling on the trade-off between performance and efficiency, highlighting that the quality and hierarchical complexity of reasoning datasets are more pivotal to performance than sheer dataset volume. Given the scarcity of high-quality reasoning datasets in the time series domain, future work could focus on expanding TSRBench to cover a broader range of domains and more complex scenarios. Additionally, VeriTime demonstrates that specialized RL training enables moderate-sized LLMs to achieve notable improvements. However, due to resource constraints, VeriTime focuses on compact models to underscore the trade-off between cost-effectiveness and performance. We foresee that applying the framework to larger foundation models could unlock greater performance gains, with improvements likely scaling positively with model size.

*Table 15.* Key hyperparameters for RFT training and inference of VeriTime.

| GRPO training | | | |
|---|---|---|---|
| max prompt length | 6190 | max response length | 3809 |
| LoRA rank | 32 | temperature | 0.8 |
| learning rate | 5e-6 | LR scheduler | Linear |
| number of generations | 4 | per-device batch size | 4 |
| Optimizer | Adamw | training epoch | 1 |
| **SFT** | | **Inference** | |
| max sequence length | 10000 | temperature | 0.2 |
| LoRA rank | 32 | | |
| learning rate | 2e-4 | | |
| LR scheduler | Linear | | |
| Optimizer | Adamw | | |
| training epoch | 1 | | |

# E. Prompt Design in TSRgen for Curating TSRBench

## E.1. TS-Tailored Chain-of-Thought

We propose a TS-tailored reasoning framework that enforces a tightly guided thought process consisting of six interdependent and logically ordered steps. All tasks in TSRBench adhere to this chain-of-thought procedure, with only minor task-specific modifications. The complete template for the inferential calculation task is provided as follows.

---

**Prompt Template for TS-Tailored Chain-of-Thought Reasoning for Inferential Calculation Task**

### Task Description
You are a time series expert. Please think step by step and strictly follow the specified output format for each step:
**Step 1. Analyzing task intent:**
Analyze the intent of the given problem and identify its core objective in the context of time series reasoning. Specifically, determine the specific object that needs to count in the numerical calculation task.
**Output Format:**
Step 1. Analyzing task intent:
[Judgment] <The specific counting object (e.g., days, occasions, significant drops). >
[Description] <Provide a rationale for the judgment based on the given problem context. Cite specific keywords or requirements from the problem and explain how these details align with the judgment. >

**Step 2. Selecting task-relevant key patterns:**
Align with the task's core objective to identify core features that directly support task completion. These features must be actionable and critical to deriving valid conclusions. Valid pattern categories include temporal patterns (e.g., trend; amplitude; fluctuation; continuity), judgment criteria (e.g., task-specific definitions of patterns), threshold values (e.g., upper bounds; lower bounds; percentage deviations), and other decisive patterns or criteria that are critical for resolving the task.
**Output Format:**
Step 2. Selecting task-relevant key patterns:
[Judgment] <Only list the names of the selected key patterns (no extra details, analysis, or conclusions); separate multiple items with semicolons. >
[Description] <For each selected pattern: Clarify the pattern's specific details; explain how it aligns with the task's core objective; elaborate on why it is critical to match the task's core objective. >

**Step 3. Analyzing time series samples using selected key patterns:**
Clarify the specific characteristics (e.g., occurrence timing, duration, intensity) and inherent rules of the selected key

---

patterns. Then evaluate whether the time series sample conforms to these task-relevant patterns and criteria. You can analyze the entire series holistically, or split it into meaningful segments (e.g., by time period, event node) based on task requirements.
**Output Format:**
Step 3. Analyzing time series samples using selected key patterns:
[Analysis] <Your analysis process. >

**Step 4. Generating preliminary answers by combining task intent and key patterns:**
Based on the analysis of task requirements, patterns, and time series data from prior steps, formulate preliminary answers.
**Output Format:**
Step 4. Generating preliminary answers by combining task intent and key patterns:
[Judgment] <Preliminary conclusion (e.g., calculated value X). >
[Description] <Provide a rationale for the judgment. >

**Step 5. Enhancing answers through reflection:**
Verify whether the selection of key patterns is comprehensive, ensuring no relevant features are omitted. Check the correctness of the analysis. Eliminate interfering factors that may affect the validity of the analysis.
**Output Format:**
Step 5. Enhancing answers through reflection:
[Analysis] <Your reflection and verification process. >

**Step 6. Summarizing the thinking process to output the answer:**
Integrate the entire analytical process, clearly presenting the complete logic from understanding the task, analyzing patterns, generating conclusions to verifying results. Finally, output an answer that meets the task requirements.
**Output Format:**
Step 6. Summarizing the thinking process to output the answer:
[Description] <Summary of the reasoning process across all steps. >
[Judgment] <Final answer.(e.g., calculated value X) (The calculated numerical value must be in digit form.) >

## E.2. Cross-LLM Validation

We present the cross-model validation template designed for advanced LLMs to evaluate the reasoning trajectories of DeepSeek-R1 across five dimensions, using a five-level rubric.

---

**Prompt Template for Cross-Model Validation**

### Task Description
You are an expert time-series analyst and reasoning auditor. Given:
– **Task type**: [Task]
– **Question**: [Question]
– **Ground-truth label**: [Label]
– **Time series data**: [Time series]
– **Teacher CoT (DeepSeek-R1)**: [Reasoning trajectory]
– **Teacher final answer**: [Final Answer]

Judge ONLY the reasoning quality of the teacher CoT. Be concise.

### Dimensions (score 1–5, integers)
1) Task-Intent Alignment: Did Step 1 correctly identify the task and required outcome?
2) Key-Pattern Relevance: Did Step 2 capture task-relevant cues/features (e.g., thresholds, trends)?
3) Logical Coherence (Steps 3–6): Are transitions and justifications consistent, with no gaps/contradictions?

---

4) Robustness vs Shortcuts: Does the reasoning avoid shortcuts/lucky guesses and cover necessary checks?

5) Bias/Artifact Check: Any unwarranted assumptions or biases unrelated to data/task?

### Rubric (per dimension)

– 5: Fully correct, precise, and justified; no hallucinations.
– 4: Minor omissions but overall correct and grounded.
– 3: Mixed quality; some unsupported or missing checks.
– 2: Major issues; key omissions or likely shortcut.
– 1: Mostly incorrect or hallucinated.

### Answer Format

Return ONLY valid JSON in this exact format:

```
{
  "scores": {
    "task_intent": 1-5,
    "key_patterns": 1-5,
    "data_grounding": 1-5,
    "logical_coherence": 1-5,
    "robustness": 1-5,
    "bias_check": 1-5
  },
  "overall_comment": "1-3 sentences, cite concrete evidence for your judgement"
}
```

