# OpenReview forum: "Time Series Reasoning via Process-Verifiable Thinking Data Synthesis and Scheduling for Tailored LLM Reasoning"
_ICML.cc/2026/Conference — ICML 2026 regular_

### Official Review · Reviewer_QoU5 · 2026-03-07

**Soundness:** 3
**Presentation:** 3
**Significance:** 3
**Originality:** 3
**Overall Recommendation:** 4
**Confidence:** 4

**Summary:**

This paper presents VeriTime, a novel framework that tailors LLMs for time series reasoning. The work is well-positioned within the emerging field of Time Series Language Models (TSLMs), and the overall direction of the paper is relevant and timely. Notably, the work focuses on advancing TSLMs that are not only trained to provide the “correct” answer, but also to evaluate and improve the structural coherence and logical consistency of the reasoning process. To support this objective, the paper introduces TSRBench, a synthetically generated text–time series reasoning dataset created through a six-step process using DeepSeek. In addition, the paper proposes a reinforcement learning approach to optimize answer correctness, reasoning accuracy, and consistency.

**Compliance With Llm Reviewing Policy:**

Affirmed.

**Final Justification:**

**Final Recommendation: Weak Accept**

This paper proposes a novel RL-based training mechanism (VeriTime) that introduces verifiable process rewards for time series reasoning. I find the idea original and potentially significant, as it targets an important limitation of existing approaches that focus only on (rewarding) final outputs rather than intermediate reasoning steps.

My main concern during the initial review was the lack of clear evidence isolating the contribution of the proposed method from confounding factors such as architecture and training data. The rebuttal has addressed this concern by providing controlled experiments across multiple architectures trained on a shared dataset, showing consistent improvements when applying VeriTime. The authors have spend a lot of effort during the short rebuttal period to generate these results. While the experimental scope still remains somewhat limited, I find the results compelling and the new experiments make the contribution more convincing from a soundness perspective.

In terms of clarity, the rebuttal also improves the positioning of the work relative to prior TSLMs and better explains the orthogonality of the proposed method.

Some weaknesses remain, including limited baseline coverage and reliance on curated process annotations, but these do not outweigh the strengths.

Overall, the rebuttal has positively changed my evaluation, and I raised my recommendation from "reject" to weak accept. I believe the general process can potentially be very impactful for the emerging field of Time Series Language Models, and the proposed method is novel and very promising.

**Key Questions For Authors:**

My main critique revolves around the novelty of this work in comparison to prior existing work.

1. The authors generate a CoT reasoning dataset for time series using a rule-based extractor with an LLM  at complex reasoning task (e.g., DeepSeek-R1) and claim:  "To the best of our knowledge, it is the first time series reasoning dataset that integrates TS-tailored CoT reasoning paths with process-verifiable annotations". How is this different from what was done by Parker et al. or OpenTSLM, which also uses reasoning LLMs to generate CoT datasets?
2. Why was a new dataset created, as there are some existing CoT datasets in the related literature? What is the advantage of TSRBench over other TSLM datasets?
3. How was the time series acutally encoded for the LLM? As a text, via learned tokens, or cross-attention?
4. Why do the experiments (Table1 and 2) not include any of the most recent Time Series Language Models (ITFormer, OpenTSLM, STReasoner, ...) for comparison?
5. Why doe the experiments in Table 2 only include VeriTime? This would be a chance to show the performance gain of your approach over already existing models.
6. Is there any novelty in the used reinforcement learning approach compared to e.g. Parker et al. or STReasoner? You mentioned "Despite these advances, existing RL-based methods in the TS domain focus primarily on optimizing final predictions, while overlooking the evaluation of reasoning processes, particularly their structural coherence and logical consistency". Could you please elaborate?

**Limitations:**

yes

**Strengths And Weaknesses:**

The novelty of this work over already existing approaches like ITFormer[1], QoQ-Med[2],OpenTSLM[3], STReasoner[4], and the approach by Parker et al. [5], is limited. The paper mainly explores approaches already introduced in recent literature, such as synthetically generated CoT time series reasoning datasets with an "expert" LLM reasoner, as well as RL approaches already explored in the mentioned papers. Additionally, this paper doesn't reference many of the mentioned papers and does not compare with them in terms of performance and novelty.

## Soundness:
Strengths:
- The paper introduces clearly the problems related to Time Series Language Models (TSLMs) such as limited datasets and limited data efficiency, rightfully acknowledging the field is still in it's infancy.
- The conducted experiments are well-designed and sound

Weaknesses:
- The gap in the related work is not clear, and much related work around TSLMs is missing. If the abovementioned papers were included, I believe the gap identified by the authors would narrow.
- The paper did not explain how exactly time series as a modality was included when training VeriTime: Was it text-tokenization, token projection, cross-attention, ..?

## Presentation
- The paper is well-written and good to follow in general
- Terms like "process-verifiable thinking" are not explained or elaborated
- Section 2.1.1 is not written "to the point". It contains many sentences, but they do not convey much concrete information and lack sufficient detail.

**For example**.
"Generating high-quality Chain-of-Thought (CoT) reasoning paths necessitates that the TS-tailored thinking strategy is both comprehensive and logically sound." What does this mean, how was this ensured?

Or: "For the key attributes extraction Step 2, TSRgen constructs a task-specific attribute set by integrating attributes identified by DeepSeek-R1 with target patterns extracted from the ChatTS dataset under a GPT selector. "
What are target patterns, what does the GPT selector do and wehre does it come from.

Or: "process-level labels are derived from expert predictions and validated against ground-truth annotations" What are process-level labels?



## Significance
- to better evaluate the results, a comparison of most recent TSLMs, such as ITFormer, OpenTSLM, Parker et al., would be beneficial; I acknowledge that a direct comparison might be challenging, however, one option would be to train VeriTime on TSRBench and then compare VeriTime against other models on TimeSeriesExam; in other words expand the evaluation on the unseen TimeSeriesExam done in the paper to include other models
- a comparison with existing TSLM datasets, esp. those containing reasoning traces, would be essential
- the improvement over existing approaches is not clear

## Originality
- Missing comparison with related work: ITFormer[1], QoQ-Med[2],OpenTSLM[3], STReasoner[4], Parker et al.[5]
- TSRBench dataset generation is not novel:  They have provided a strong reasoning LLM with a time series and ask it to reason about it in a six-step process. Similar approaches have been done by the aforementioned papers.
- RL approach is not entirely novel and more like an expansion of methods tried in prior work, see STReasoner, Parker et al.


[1] Wang, Y., Lei, P., Song, J., Hao, Y., Chen, T., Zhang, Y., Jia, L., Li, Y., & Wei, Z. (2025). ITFormer: Bridging time series and natural language for multi-modal QA with large-scale multitask dataset.

[2] Dai, W., Chen, P., Ekbote, C., & Liang, P. P. (2025). QoQ-Med: Building multimodal clinical foundation models with domain-aware GRPO training.

[3] Langer, P., Kaar, T., Rosenblattl, M., Xu, M. A., Chow, W., Maritsch, M., Jakob, R., Wang, N., Liu, J., Verma, A., Han, B., Kim, D. S., Chubb, H., Ceresnak, S., Zahedivash, A., Sandhu, A. T. S., Rodriguez, F., McDuff, D., Fleisch, E., Aalami, O., Barata, F., & Schmiedmayer, P. (2026). OpenTSLM: Time-series language models for reasoning over multivariate medical text- and time-series data

[4] Ni, J., Wang, S., Jin, M., He, Q., & Jin, W. (2026). STReasoner: Empowering LLMs for spatio-temporal reasoning in time series via spatial-aware reinforcement learning.

[5] Parker, F., Chan, N., Zhang, C., & Ghobadi, K. (2025). Augmenting LLMs for general time series understanding and prediction

---

> ### Author Rebuttal · Authors · 2026-03-31
>
> > #### W1. Research Gap & Related Work
>
> The primary contributions of these works lie in aspects different from ours.
>
> i) TsLLM, OpenTSLM, and ITFormer focus on time series (TS) data encoding and fusing the TS modality with LLMs by training the encoding and fusion modules (i.e., the modules that generate embeddings **prior to the TS embedding is passed to LLM**) via supervised fine-tuning on either labeled datasets without Chain-of-Thought (CoT) or offline CoT data.
>
> ii) STReasoner was first posted on arXiv in January 2026, shortly before the ICML deadline, and focuses on the related but distinct spatiotemporal domain.
>
> iii) QoQ-Med targets the medical domain rather than time series.
>
> In contrast, our work focuses on enhancing LLMs' TS-dedicated reasoning capability through on-policy RL **after TS data is passed to the LLM**. In particular, we are the first to introduce verifiable process rewards to address the unique challenges of time series data that are often complex, multi-scale temporal dynamics. The multi-step reasoning potential of LLMs is constrained by the scarcity of process-labeled TS reasoning data and the lack of RL algorithms designed to exploit such information. The table below summarizes the key distinctions.
>
> ||Domain|Training|RL|Process Verifiation|
> |-|-|-|-|-|
> |ITFormer|Aeroengine|SFT|×|×|
> |OpenTSLM|Medical|SFT|×|×|
> |QoQ-Med|Medical (Visual)|SFT+RL|√|×|
> |TsLLM|General|Pretraining+SFT|×|×|
> |STReasoner|Spatio-Temporal|SFT+RL|√|×|
> |VeriTime|General|SFT+RL|√|√|
>
> The main contributions of these works lie in orthogonal aspects to ours, which can in future be integrated with ours by integrating their encoding/fusion modules into end-to-end training with the LLM under our VeriTime RL algorithms on TSRBench. We will clarify the comparisons and discuss potential integration with improved encoding/fusion strategies as future work.
>
> ---
> > #### W2&Q3: Time Series Encoding
>
> We focus on improving LLMs’ reasoning capability **after** the time series data has entered the LLM, and therefore adopt a simple encoding strategy by directly representing the data as text. As discussed in W1 above, we anticipate that leveraging advanced encoding and fusion or even integrating the encoding/fusion modules into VeriTime’s training pipeline could further enhance VeriTime’s performance.
>
> > #### Q4&W1: Baselines on TimeSeriesExam
>
> We conduct additional experiments to compare with the requested baselines, i.e., STReasoner, ITFormer, and OpenTSLM, on the TimeSeriesExam dataset. TsLLM is currently not open-sourced, and QoQ-Med focuses exclusively on the clinical visual domain. ITFormer and OpenTSLM underperform to VeriTime, likely due to their reliance on domain-specific training corpora (aeroengine and medical) and the absence of explicit reasoning mechanisms.
>
> |VeriTime|STReasoner|ITFormer|OpenTSLM|
> |-|-|-|-|
> |47.27|44.96|36.07|32.69|
>
> ---
> > #### Q5: Baselines on Table 2
>
> We have included comparisons between VeriTime against classical models, general LLMs, and TSLMs in Table 2 in our original manuscript. VeriTime demonstrates clear performance gains over these baselines.
>
> ---
> > #### Q1&Q2&Q6: Regarding Novelty Compared to Existing Reasoning Benchmarks & RL Approaches
>
> Thank you for raising this important point.
>
> i) Comparison with SFT-on-CoT-based Reasoning Methods. The first category relies primarily on Supervised Fine-Tuning (SFT) using CoT data, e.g., TsLLM (Parker et al.). They first synthesize offline CoT trajectories and optimize the model using supervised learning losses, e.g., cross-entropy or MSE. However, relying solely on offline CoT (meaning that reasoning trajectories remain fixed despite updates to the model under training) is known to limit exploration and cap reasoning potential [2]. In contrast, VeriTime adopts an on-policy RL training paradigm based on GRPO, which effectively mitigates these limitations by allowing the model to actively explore the reasoning space and continuously improve by generating reasoning trajectories during training (rather than fixed pretraining) [3].
>
> ii) Comparison with Outcome-Reward-Only RL Reasoning Methods. This category relies on outcome-reward-only on-policy RL, e.g., STReasoner in the spatiotemporal domain and RL-based approaches in the time series domain [1]. These methods treat only the final output as rewards. However, TS data often exhibits complex, multi-scale temporal dynamics, and the multi-step reasoning potential of LLMs is limited by the scarcity of process-labeled TS reasoning data and the lack of RL algorithms tailored to exploit such information. To address this, VeriTime introduces verifiable process rewards that explicitly evaluate intermediate reasoning steps .
>
> [1] *Eliciting Chain-of-Thought Reasoning for Time Series Analysis using Reinforcement Learning (Arxiv Oct 2025)*
>
> [2] *Training language models to follow instructions with human feedback (Neurips22)*
>
> [3] *DeepSeek-R1: Incentivizing Reasoning Capability in LLMs via Reinforcement Learning*

---

> > ### Author Rebuttal · Reviewer_QoU5 · 2026-04-02
> >
> > Thank you very much for providing the additional details and experiments. They address several of my concerns.
> >
> > In general, I agree that VeriTime can be seen as somewhat orthogonal to concrete TSLMs such as ITFormer, STReasoner, and OpenTSLM (fyi: QoQ-Med also does have a time series encoder and would be suitable for comparison). The mechanism introduced by VeriTime could, in principle, enable all of these models to achieve improved performance. In particular, the idea of rewarding both the final answer and intermediate reasoning steps is novel and tackles an important and challenging problem.
> >
> > However, what I am currently missing is a clear way to quantify the specific contribution of VeriTime itself. While the authors added comparisons between VeriTime and related models, and VeriTime achieves the best performance, I believe that this comparison is not entirely fair.
> >
> > Since VeriTime is fundamentally a training mechanism for TSLMs, a more appropriate evaluation would be to isolate its effect. A clear way to demonstrate this would be to (1) train the related models on a common dataset such as TimeSeriesExam (or another suitable benchmark), and compare this to (2) training the same models on the same data but using the VeriTime mechanism.
> >
> > In other words, it would be important to disentangle performance differences arising from model architectures and training data (I believe the authors have not trained ITFormer on TimeSeriesExam for their comparison), from those stemming from the improved training procedure introduced by VeriTime.
> >
> > If such an experiment can demonstrate consistent gains attributable to VeriTime, I would be happy to consider raising my score.

---

> > > ### Author Response · Authors · 2026-04-07
> > >
> > > We sincerely thank the reviewer for the constructive suggestion. We truely apologize for the delay in our follow-up response. We began running new experiments immediately upon seeing your comment, though reproducing additional baselines requires substantial implementation effort, including resolving framework compatibility issues, understanding the original implementation details of each model, and the time needed for training. It has taken us several days, and the interactive rebuttal period is now drawing to a close. Given these constraints, we were able to reproduce and evaluate two baseline training algorithms across three model architectures, and we describe our procedure in detail below.
> > >
> > > **Training Dataset**.  Following your suggestion, we use a common dataset, ChatTS, as the shared training corpus. We have to opt for this dataset for two reasons. First, it allows us to reserve TimeSeriesExam for evaluation, which allows the results to better reflect performance on unseen data. Second, annotating process-verifiable labels and reasoning trajectories for a new dataset such as TimeSeriesExam is challenging within the rebuttal period, whereas ChatTS has already been curated with such annotations as part of our TSRBench pipeline, making it a practical choice under this situation.
> > >
> > > **ChatTS Dataset Details.** ChatTS is a dataset pairing synthetic time series with textual descriptions, where each sample is a question-answer pair without Chain-of-Thought reasoning trajectories or process-reward annotations. During TSRBench curation, we filter ChatTS into three task categories, i.e., Anomaly Detection, Scenario Attribution, and Inferential Calculation, retaining only samples with sufficient reasoning complexity and annotation quality to serve as meaningful training signal. All baselines and VeriTime are trained on this same filtered subset to ensure a fair comparison.
> > >
> > > **Baseline Training Algorithm**. During the rebuttal period, we successfully replicate two representative baseline training algorithms, ITFormer [2] and OpenTSLM [3], each trained following their original procedures. For ITFormer, we first pre-train the time series encoder on ChatTS, then perform end-to-end supervised fine-tuning (SFT) that jointly bridges the encoder with the LLM backbone. For OpenTSLM, we follow its curriculum learning protocol and train the patch encoder and LLM via LoRA adapters on ChatTS.
> > >
> > > **VeriTime Training Algorithm**. When applying VeriTime to each baseline architecture, we freeze the respective time series encoder and train only the LLM component — first via SFT on our process-verifiable CoT trajectories, followed by our process-verifiable RL training stage. This design deliberately isolates reasoning improvement to the language modeling component and ensures a controlled, architecture-neutral comparison.
> > >
> > > **Evaluation Results**. We evaluate the models trained by respective training algorithms on the held-out TimeSeriesExam with the table shown below.
> > >
> > > |Model Arthitectures|OpenTSLM|ITFormer-0.5B|ITFormer-3B|
> > > |-|-|-|-|
> > > |Base Training Algorithms|28.12%|33.33%|36.43%|
> > > |+VeriTime|46.22%|44.96%|44.03%|
> > >
> > > VeriTime yields consistent and substantial gains across all tested TSLM architectures, e.g., a +18.10%, +11.63%, and +7.60% improvement for OpenTSLM, ITFormer-0.5B, and ITFormer-3B, respectively, which demonstrate that process-level reward supervision meaningfully and reliably enhances time series reasoning, independent of the underlying model architecture.
> > >
> > > **Planned Revision**. In our revised paper, we will additionally split TimeSeriesExam into training and test partitions, annotate the training split with process-verifiable labels using our TSRGen pipeline, and conduct a fully in-domain controlled comparison on the test split, providing the most rigorous possible evaluation of VeriTime's contribution. Furthermore, we will extend our replication efforts to include all remaining baselines suggested by the reviewer that could not be completed within the rebuttal period.
> > >
> > > We hope these controlled experiments, i.e., holding both architecture and training data constant, directly address the reviewer's core concern and provide compelling evidence that the gains are attributable to the VeriTime mechanism itself. Once again, we sincerely thank you for recognizing the novelty of rewarding both the final answer and intermediate reasoning steps as an important and challenging contribution. We appreciate your time and thoughtful comments, and we would be grateful if you would consider raising your score in light of these new results.
> > >
> > > &nbsp;
> > >
> > > **References**
> > >
> > > #### [1] *ChatTS: Aligning Time Series with LLMs via Synthetic Data for Enhanced Understanding and Reasoning (VLDB25)*
> > >
> > > #### [2] *ITFormer: Bridging time series and natural language for multi-modal QA with large-scale multitask dataset (ICML25)*
> > >
> > > #### [3] *OpenTSLM: Time-Series Language Models for Reasoning over Multivariate Medical Text- and Time-Series Data (Arxiv Oct 2025)*

---

### Official Review · Reviewer_o2zP · 2026-03-08

**Soundness:** 2
**Presentation:** 2
**Significance:** 2
**Originality:** 2
**Overall Recommendation:** 3
**Confidence:** 4

**Summary:**

This paper proposes VeriTime, a framework for improving time-series reasoning in LLMs through three components: a synthetic data pipeline (TSRgen) for building a time-series reasoning dataset with process-verifiable annotations, a difficulty-aware data scheduling strategy, and a two-stage training pipeline combining SFT and GRPO-based RL. The resulting dataset, TSRBench, covers both scenario-based and knowledge-based tasks across seven task types, mixing synthetic and real-world time series. The method uses a TS-tailored six-step CoT template and designs multi-objective rewards for task comprehension, critical pattern selection, answer alignment, answer verification, structure, and output length.

**Compliance With Llm Reviewing Policy:**

Affirmed.

**Final Justification:**

Thanks for the response. I still think the novelty and potential impact of this paper is limited.

After reviewing all the reviews, I decide to keep my score unchanged.

**Key Questions For Authors:**

see weaknesses

**Limitations:**

see weaknesses

**Strengths And Weaknesses:**

Strength

1.	Time-series reasoning with LLMs is still underexplored, and the paper addresses an important gap by focusing not only on final-answer supervision but also on intermediate reasoning signals.

2.	The paper evaluates on scenario-based tasks, knowledge-based tasks, TimeSeriesExam, and DROP, and also includes ablations on reward components and data scheduling.

Weaknesses:

1.	The novelty is limited. The paper combines synthetic data generation, a task-specific CoT template, curriculum-like data scheduling, and RL fine-tuning with shaped rewards. While the full pipeline is reasonably complete, none of these components is novel, and the paper does not clearly isolate what the main conceptual novelty is.

This makes the work feel more like a system integration paper than a method with a sharply defined new idea.

2.	The paper does not provide training results on models beyond 7B scale, and the main experiments are conducted only on compact 3B–4B models. As a result, the effectiveness of the method is not yet fully validated at larger model scales.

3.	The benchmark lacks top model evaluation such as GPT-5, Gemini 3-Pro. Therefore, it remains unclear how strong the proposed framework is relative to the current frontier of reasoning-capable LLMs.

---

> ### Author Rebuttal · Authors · 2026-03-31
>
> > #### W1: Main Novelty
>
> We sincerely thank you for raising this important point. We agree that components such as RL and CoT are well-explored in **general domains**. In the time series (TS) domain, however, reasoning remains under-explored and presents TS-specific challenges that demand tailor-made algorithmic designs. Existing TS architectures are largely confined to forecasting and lack generalized reasoning capabilities, as also highlighted by emerging results such as TimeOmni [1], which was accepted at ICLR'26 and open-sourced its code and model after the ICML submission deadline on February 7, 2026.
>
> Beyond [1], which has not yet addressed process verifiability, our work further notes that TS data inherently involves complex multi-scale temporal dynamics. Unlocking the multi-step reasoning potential of LLMs is therefore constrained by the scarcity of process-labeled TS reasoning data and the lack of RL algorithms specifically designed to leverage such data. Consequently, developing dedicated TS reasoning models requires domain-specific, nontrivial technical innovations.
>
> Motivated by this, our conceptual novelty lies in introducing the first process-verifiable reasoning framework tailored for time series. We design TS-tailored CoT trajectories paired with fine-grained, multi-objective process rewards. This mechanism uniquely isolates and supervises the logical validity of intermediate temporal reasoning steps, rather than merely evaluating final predictions. As shown in the table below, the performance of VeriTime compared to TimeOmni and other baselines on the same test split of knowledge-based tasks further demonstrates that our process-level supervision provides an effective methodological advancement. We will incorporate the above clarifications and new empirical results in the revised paper.
>
> |Acc (%)|CTU|ECG|EMG|RCW|
> |-|-|-|-|-|
> |VeriTime (Qwen3-4B)|66.67|33.33|71.23|63.27|
> |TimeOmni|50.00|26.26|34.25|55.10|
> |Time-R1 [2]|59.17|23.74|42.86|33.52|
> |Time-R1 [3]|43.40|17.17|34.72|32.61|
>
> [1] *TimeOmni-1: Incentivizing Complex Reasoning with Time Series in Large Language Models (ICLR26)*
>
> [2] *Time Series Forecasting as Reasoning: A Slow-Thinking Approach with Reinforced LLMs (Arxiv June 2025)*
>
> [3] *Time-R1: Towards Comprehensive Temporal Reasoning in LLMs (Arxiv May 2025)*
>
> ---
> > #### W2: Training Results on Models Beyond 7B Scale
>
> Thank you for your valuable feedback. To validate the effectiveness of VeriTime training on larger model scales according to your suggestion, we conduct additional experiments using Qwen3-8B and Qwen2.5-14B. As shown in the table below, VeriTime can generalize to larger model sizes,  achieving performance improvements over the respective base models. Moreover, VeriTime generally attains better results with increasing model size. These findings demonstrate that the benefits of the process-verifiable reasoning approach scale effectively beyond the compact 3B–4B regime. Due to the limited time available during the rebuttal period, we did not conduct sufficient hyperparameter tuning and will incorporate these new evaluations in the revised paper.
>
> |Acc (%)|CTU|ECG|EMG|RCW|
> |-|-|-|-|-|
> |Base (Qwen3-8B)|48.33|24.24|47.95|46.94|
> |VeriTime (Qwen3-8B)|71.67|31.21|65.96|58.16|
> |Base (Qwen2.5-14B)|58.33|23.71|38.36|47.96|
> |VeriTime (Qwen2.5-14B)|81.67|33.33|72.22|67.35|
>
> ---
> > #### W3: Regarding Top Model Evaluation such as GPT-5, Gemini 3-Pro
>
> Thank you for the suggestion to evaluate frontier reasoning models to better contextualize VeriTime's performance. We have conducted the requested evaluations on GPT-5 and Gemini 3-Pro, as summarized in the table below. Due to the time constraints of the rebuttal period, we focus evaluations on the four more challenging knowledge-based tasks.
>
> |Acc (%)|CTU|ECG|EMG|RCW|
> |-|-|-|-|-|
> |VeriTime (Qwen3-4B)|66.67|33.33|71.23|63.27|
> |GPT-5|50.00|25.00|58.33|53.64|
> |Gemini-3-Pro|50.85|28.38|61.64|65.31|
> |VeriTime (Qwen2.5-14B)|81.67|33.33|72.22|67.35|
>
> In general, VeriTime outperforms these reasoning-capable LLMs, as the latter are not explicitly tailored for time series reasoning. We do note that Gemini-3-Pro marginally exceeds VeriTime on the RCW dataset by 2%, which is understandable given the substantial difference in model size (VeriTime is based on a 4B model). We plan to conduct further evaluations on additional datasets and tasks that could not be included due to the limited time of the rebuttal period, and we will incorporate these new results in the revised paper according to your suggestions.

---

> > ### Author Rebuttal · Reviewer_o2zP · 2026-04-03
> >
> > Thanks for the response. I still think the novelty and potential impact of this paper is limited.
> >
> > After reviewing all the reviews, I decide to keep my score unchanged.

---

### Official Review · Reviewer_BGrZ · 2026-03-09

**Soundness:** 3
**Presentation:** 2
**Significance:** 2
**Originality:** 3
**Overall Recommendation:** 4
**Confidence:** 5

**Summary:**

This paper introduces VeriTime. It is a framework designed to enhance the time series reasoning capabilities of LLMs through the automated synthesis of a process-verifiable CoT dataset, a difficulty-aware data scheduling mechanism, and a two-stage reinforcement learning fine-tuning strategy guided by multi-objective process rewards.

**Compliance With Llm Reviewing Policy:**

Affirmed.

**Key Questions For Authors:**

No further questions

**Limitations:**

yes

**Strengths And Weaknesses:**

S1. The transition from outcome based rewards to explicitly process-verifiable rewards in the time series domain is novel.

S2. The empirical results are compelling, demonstrating that the VeriTime can elevate the reasoning capabilities of compact 3B and 4B parameter models.

S3. The ablation studies in experiment sections isolate the contributions of the individual reward components, generation settings, and data scheduling strategies, providing an evidence for the necessity of each architectural choice

W1. The synthesis pipeline relies heavily on teacher model DeepSeek-R1 and rule-based extractors to generate the reasoning trajectories and "process-verifiable" labels, and the paper uses cross-model validation (llm-as-judge), which could suffer from known biases (as today`s language model are trained on similar data).

W2. The lack of human-annotated ground truth to verify these intermediate reasoning steps raises concerns about hallucinations or flawed reasoning logic being baked into the TSRBench dataset.

W3. While the ablation studies are strong, the main baselines consist only of zero-shot base LLMs and basic SFT time-series models. To truly prove the superiority of the tailored RL approach, the authors should compare VeriTime against a baseline trained with RL, like TimeMaster [1]

[1] TimeMaster: Training Time-Series Multimodal LLMs to Reason via Reinforcement Learning.

(S for Strengths, and W for Weaknesses)

---

> ### Author Rebuttal · Authors · 2026-03-31
>
> > #### W1: Process-Verifiable Labels
>
> We sincerely thank you for raising these important points regarding potential biases in LLM judgments and the insufficiency of human verification. To assess the robustness of DeepSeek-R1 as the teacher model, we test 50 samples on select Gemini3-pro and GPT-4o-mini using our tailored CoT template. The resulting reasoning trajectories show high semantic similarity (0.93 for DeepSeek vs. Gemini, 0.91 vs. GPT), proving transferability of the framework. Manual inspection further finds that DeepSeek-R1 and Gemini-3 follow nearly identical logical paths. DeepSeek-R1 also produces more concise and precise reasoning, while GPT-4o-mini is less effective at capturing exact temporal intervals and noise morphology. In future work, we plan to explore aggregating trajectories from multiple LLMs to enrich TSRBench.
>
> To verify the pattern relevance of TSRBench, we train a dedicated classifier SVM to identify typical pattern characteristics and compare its predictions with those of our teacher model, DeepSeek-R1. As shown in the table, the high agreement rates (%) validate that DeepSeek's inherent perception consistently aligns with a supervised baseline.
>
> ||Amplitude|Trend|Threshold|Spike|
> |-|-|-|-|-|
> |SVM|54.58|20.42|43.75|20.00|
> |DeepSeek-R1|53.33|24.58|43.75|23.33|
> |Agreement Rate|85.42|90.83|91.67|57.50|
>
> Besides, to verify intent alignment, we utilize a regular expression matching tool to assign a similarity score (from 1 to 5) and compare it against evaluations from frontier LLMs. The results show that the objective tool assessment aligns closely with the LLM judgments.
>
> ||Matching Tool|GPT-5|Gemini|Qwen3-235B|
> |-|-|-|-|-|
> |Score|4.72|4.97|5.00|4.99|
>
> Relying on an LLM as a judge involves an inherent trade-off between scalability and precision. In future work, we will discuss this dynamic and explore human-in-the-loop methodologies to better balance efficient human verification with large-scale automated scoring.
>
> > #### W2: Human Annotation
>
> Thank you for highlighting the importance of human verification. To ensure quality, we manually audited about **45%** of the process labels to remove invalid values generated by the TSRgen pipeline. Unlike natural language or vision data, time series lacks intuitive semantics, making manual pattern annotation inherently complex and hard to scale. To bridge this gap, our automated pipeline provides an effective solution for large-scale data generation.
>
> ---
> > #### W3: RL baseline
>
> We compare VeriTime with the suggested TimeMaster, as well as three other recent RL-based baselines that do not use image modalities. Their methodological configurations are detailed in the first table below. We select these methods as RL baselines because they represent the current frontier of RL applied to time series. In particular, TimeOmni was recently accepted at ICLR-26 and open-sourced after the ICML submission deadline on February 7, 2026, making us unable to directly compare it at the time of submission. A key distinction is that TimeMaster incorporates image modalities, whereas the other three baselines and VeriTime focus on time series and text modalities, each employing different RL algorithm designs. This setup allows us to directly contrast existing approaches, which primarily optimize for format compliance and final prediction accuracy, with our proposed process-verifiable framework.
>
> ||Modality Setting|RL Reward Design|
> |-|-|-|
> |TimeMaster|Image + Numeric TS + Text|Format, Prediction Accuracy, LLM-as-a-Judge (for extension content)|
> |Time-R1 [1]|Numeric TS + Text|Format, Prediction Accuracy, Structural Similarity|
> |Time-R1 [2]|TS-related Text|Format, Prediction Accuracy, Negative Penalty|
> |TimeOmni [3]|Numeric TS + Text|Format, Prediction Accuracy|
>
> As shown in the performance table below, VeriTime demonstrates better results across most datasets. Although TimeMaster achieves a higher score on the RCW dataset, this is likely because it incorporates visual modalities for reasoning enhancement. Overall, these results validate that our process-verifiable framework provides more effective supervision than standard or outcome-based RL approaches. In future work, we plan to extend VeriTime to incorporate image modalities using multimodal LLMs, similar to TimeMaster, which we anticipate will further enhance its performance. We will incorporate the new comparison results, along with the discussion of future work, in the revised paper.
>
> |Acc (%)|CTU|ECG|EMG|RCW|
> |-|-|-|-|-|
> |VeriTime (Qwen3-4B)|66.67|33.33|71.23|63.27|
> |TimeMaster|54.00|25.00|48.78|72.53|
> |Time-R1 [1]|59.17|23.74|42.86|33.52|
> |Time-R1 [2]|43.40|17.17|34.72|32.61|
> |TimeOmni [3]|50.00|26.26|34.25|55.10|
>
> [1] *Time Series Forecasting as Reasoning: A Slow-Thinking Approach with Reinforced LLMs (Arxiv June 2025)*
>
> [2] *Time-R1: Towards Comprehensive Temporal Reasoning in LLMs (Arxiv May 2025)*
>
> [3] *TimeOmni-1: Incentivizing Complex Reasoning with Time Series in Large Language Models (ICLR26)*

---

> > ### Author Rebuttal · Reviewer_BGrZ · 2026-04-07
> >
> > sorry, I read your response earlier but forgot to reply, and thank you for the additional clarifications.
> >
> > I believe this paper is acceptable, but I will maintain my current score 4.

---

### Official Review · Reviewer_3BAU · 2026-03-11

**Soundness:** 3
**Presentation:** 4
**Significance:** 3
**Originality:** 3
**Overall Recommendation:** 4
**Confidence:** 4

**Summary:**

VeriTime is a framework for enhancing time series reasoning in large language models, built on three key components: a synthesized time series reasoning dataset with process-verifiable annotations (TSRBench via TSRgen), a difficulty-based data scheduling strategy, and a two-stage fine-tuning pipeline combining SFT and RL. The method devises a TS-tailored six-step reasoning format and multi-objective rewards intended to supervise both final answers and intermediate reasoning steps. Experiments on scenario-based and knowledge-based time series tasks, plus TimeSeriesExam and DROP, show sizable gains over the Qwen2.5-3B and Qwen3-4B models.

**Compliance With Llm Reviewing Policy:**

Affirmed.

**Final Justification:**

I maintain my rating. In their rebuttal, the authors provided comprehensive TSRBench statistical data and SFT ablation studies, which effectively clarified concerns regarding experimental and dataset details. However, I remain concerned about the robustness of the automated process label generation and its long-term impact on the model's reasoning capabilities. Nevertheless, this paper presents a technically sound framework for logical reasoning over time series and represents a valuable contribution. My current score accurately reflects both the merits and limitations of this work.

**Key Questions For Authors:**

1. What are the exact sizes and splits of TSRBench by task and domain, and what fraction of process labels were audited?
2. What are the TSRBench-SFT-only results on the knowledge-based tasks in Table 2 and on the transfer tasks in Table 3?
This is the cleanest missing ablation for understanding whether RL is actually carrying its claimed share of the improvement.

**Limitations:**

yes

**Strengths And Weaknesses:**

**Strengths**

- The paper tackles a relevant and current challenge. While LLMs are often used for text-based reasoning, applying them to time-series data is a less explored area. This work goes beyond treating it as simple number-crunching and approaches it as a genuine reasoning task.
- The methodology is clearly structured. Figure 1 and Figure 2 outline the process in a straightforward way: generating training data, using a progressive learning schedule, and applying reinforcement learning with rewards tied to each step.
- The performance improvements are substantial. In Table 1, the Qwen2.5-3B model jumps from 40.99 to 82.86 on scenario-based tasks. Table 2 also shows clear gains on benchmarks like CTU, EMG, and RCW.

**Weaknesses**

- The paper’s new benchmark is not sufficiently documented. Although the title and abstract highlight “process-verifiable thinking data synthesis,” the main text does not clearly summarize the dataset. Missing details include: example counts, splits across domains or into train/validation/test sets, label distribution, and how many examples actually have step-by-step labels.
- The "process-verifiable" reasoning framework is not robust. Section 2.1.3 indicates that step labels are generated by comparing outputs from multiple LLMs via simple rules, without evaluating key steps like critical segment analysis or self-reflection. This results in only partial verifiability, heavily dependent on the label-generation process. If the labels are noisy or unreliable, reinforcement learning may merely train the model to imitate artifacts of the labeling method, rather than learn truly sound reasoning.

---

> ### Author Rebuttal · Authors · 2026-03-31
>
> > #### W1&Q1: TSRBench Statistics
>
> We sincerely appreciate your valuable feedback. We provide a complete summary of TSRBench, covering all requested aspects.
>
> i) Example Counts and average time points for each task and dataset are:
> |Task|Anomaly Detection|Scenario Attribution|Inferential Calculation|CTU|ECG|EMG|RCW|
> |-|-|-|-|-|-|-|-|
> |Example Counts|1180|930|410|270|780|450|320|
> |Avg. Time Points|300|300|324|720|500|600|500|
>
> ii) Split Ratio across training and test sets is approximately 5:1.
>
> iii) Label Distribution. The final labels are formatted as True/False (anomaly detection), open-ended QA (inferential calculation), or multiple-choice (others). TSRBench provides diverse intermediate reasoning signals, comprising 9,746 occurrences across 2,684 unique pattern labels. The top five most frequent patterns are detailed in the table below.
>
> |Amplitude|Fluctuation|Continuity|Trend|Upward Spike|
> |-|-|-|-|-|
> |5.78%|4.99%|4.72%|3.87%|2.43%|
>
> iv) Process Labels Coverage. We have rigorously filtered all samples for final-answer correctness and manually audited about 45% of the process labels to remove invalid values, **All examples** in TSRBench are equipped with complete step-by-step labels.
>
> ---
> > #### W2: Regarding Verifiability of Process-Verifiable Labels
>
> We thank the reviewer for this valuable comment. We provide additional empirical evidence to investigate the reliability of the reasoning framework.
>
> i) Overall Reasoning Quality of Reasoning Steps 3-6. We conduct a cross-model evaluation of the intermediate reasoning steps 3–6 produced by the teacher model. We prompt frontier LLMs to score the reasoning quality on a 1-to-5 scale across three dimensions: Logical Coherence, Robustness, and Bias Check. As shown below, the consistently high scores indicate strong inter-judge agreement and provide quantitative evidence for the reliability of the intermediate reasoning paths. Please also refer to Appendix C.2 for detailed LLM scoring results and Appendix F.2 for cross-model validation prompt templates.
>
> |Score|Logical Coherence|Robustness|Bias Check|
> |-|-|-|-|
> |GPT-5|4.50|4.30|4.62|
> |Gemini|4.31|4.38|4.62|
> |Qwen3-235B|4.70|4.56|4.70|
>
> ii) Reasoning Quality of Critical Segment Analysis and Self-Reflection. We assess their reasoning quality based on two specific dimensions: (1) Verifying whether the model accurately isolates and analyzes essential data segments without hallucinating values or patterns, and (2) Evaluating whether the reasoning included explicit verification steps to cross-check initial findings against raw data and task constraints. The average scores across all evaluating LLMs consistently exceeded 4.0 in the range 1-5, providing additional evidence that these two reasoning steps are relatively reliable.
>
> |Score|Critical Segment Analysis|Self-Reflection|
> |-|-|-|
> |GPT-5|4.50|4.30|
> |Gemini|4.12|4.11|
> |Qwen3-235B|4.72|4.84|
>
> iii) Consistency of Intent Recognition. We utilize a regular expression matching tool to assign an objective similarity score (from 1 to 5). The results show that the assessment aligns closely with the LLM judgments.
>
> ||Matching Tool|GPT-5|Gemini|Qwen3-235B|
> |-|-|-|-|-|
> |Score|4.72|4.97|5.00|4.99|
>
> iv) Reliability of Process Labels. We evaluate the temporal pattern perception accuracy of the LLM. We train a dedicated classifier (SVM) to identify temporal patterns across 300 synthetic test examples and compare its classifications with the LLM's judgments. As shown below, the high agreement rates (%) indicate that the LLM's temporal perception aligns closely with that of a dedicated classifier tailored for temporal pattern classification.
>
> ||Amplitude|Trend|Threshold|Spike|
> |-|-|-|-|-|
> |SVM|54.58|20.42|43.75|20.00|
> |DeepSeek-R1|53.33|24.58|43.75|23.33|
> |Agreement Rate|85.42|90.83|91.67|57.50|
>
> These results suggest current reasoning framework produces key reasoning steps of relatively high quality, annotates process labels that align with dedicated classifier, and exhibits strong cross-model consistency among SOTA LLMs. That being said, we fully agree with your suggestion that enabling more rigorously verifiable reasoning is an important future direction, e.g., studying agentic reasoning to call tools/retrieval/search for precise temporal pattern analysis and fact checking, or explicitly incorporating human-in-the-loop to involve domain experts to verify the annotation. We will include these future work discussions.
>
> ---
> > #### Q2: TSRBench-SFT Ablation Results
>
> We conduct SFT-only ablation experiments using Qwen3-4B, which confirms RL's crucial role in enhancing time series reasoning. The SFT serves the goal of teaching the model to follow the designed reasoning format. The average format reward score (on a 3-point scale) for SFT-only is 2.74, i.e., a substantial improvement over the base model with a score close to 0.
>
> |Acc (%)|CTU|ECG|EMG|RCW|TimeSeriesExam|DROP|
> |-|-|-|-|-|-|-|
> |SFT-only|62.50|23.23|55.78|36.92|40.31|69.80|
> |VeriTime|67.50|30.30|65.31|65.39|47.27|80.00|

---

> > ### Author Rebuttal · Reviewer_3BAU · 2026-04-04
> >
> > Thanks for the detailed response and for providing the additional ablation studies and dataset statistics. The new results on TSRBench-SFT and the audit of process labels help clarify the contribution. However, I still have some reservations about the robustness of the automatic label generation process and its long-term impact on model reasoning. Therefore, I believe the current score (4: Weak Accept) accurately reflects the paper's contribution and its remaining limitations. I will keep my score unchanged.

---

### Decision · Program_Chairs · 2026-04-30

**Decision:**

Accept (regular)

**Comment:**

This paper introduces VeriTime, a framework for improving time series reasoning in language models through synthesized process-verifiable reasoning data, difficulty-aware data scheduling, and a two-stage fine-tuning pipeline with reinforcement learning. The reviewers found the problem timely and relevant, and noted that the paper presents a technically solid framework with substantial empirical gains on a range of time series reasoning benchmarks.

During the rebuttal and discussion period, the reviewers discussed both the strengths and the remaining limitations of the work. Reviewers agreed that the rebuttal clarified important experimental and dataset details, particularly by providing additional TSRBench statistics and ablation results. Several reviewers found that the controlled experiments better supported the claim that the gains come from the proposed training mechanism rather than architecture or data differences alone, and viewed the process-reward formulation for time series reasoning as a meaningful contribution that could be useful beyond a single model instantiation. At the same time, some concerns remained regarding the degree of methodological novelty, the robustness of the automatically generated process labels, the paper’s positioning relative to recent related work, and the need for stronger comparisons to additional baselines. Overall, however, the discussion suggests that the paper is technically sound and that its empirical contributions and practical value are sufficient for acceptance, even if some of the framing and evaluation could be further strengthened.

In light of these feedback, we recommend weak acceptance. The paper has some clear limitations, but the reviewers found that it makes a useful contribution to an active area and is likely to be of interest to the community.